# Selective increases in inter-individual variability in response to environmental enrichment in female mice

Julia C Körholz[1,2†], Sara Zocher[1,2†], Anna N Grzyb[1,2†], Benjamin Morisse[1,2], Alexandra Poetzsch[1,2], Fanny Ehret[1,2], Christopher Schmied[1,2], Gerd Kempermann[1,2]*

[1]German Center for Neurodegenerative Diseases (DZNE) Dresden, Dresden, Germany; [2]CRTD – Center for Regenerative Therapies Dresden, Technische Universität Dresden, Dresden, Germany

**Abstract** One manifestation of individualization is a progressively differential response of individuals to the non-shared components of the same environment. Individualization has practical implications in the clinical setting, where subtle differences between patients are often decisive for the success of an intervention, yet there has been no suitable animal model to study its underlying biological mechanisms. Here we show that enriched environment (ENR) can serve as a model of brain individualization. We kept 40 isogenic female C57BL/6JRj mice for 3 months in ENR and compared these mice to an equally sized group of standard-housed control animals, looking at the effects on a wide range of phenotypes in terms of both means and variances. Although ENR influenced multiple parameters and restructured correlation patterns between them, it only increased differences among individuals in traits related to brain and behavior (adult hippocampal neurogenesis, motor cortex thickness, open field and object exploration), in agreement with the hypothesis of a specific activity-dependent development of brain individuality.
DOI: https://doi.org/10.7554/eLife.35690.001

*For correspondence:
gerd.kempermann@dzne.de

†These authors contributed equally to this work

Competing interests: The authors declare that no competing interests exist.

## Introduction

Individualization is the process of developing unique traits and thus divergence from the inborn and genetically determined makeup. The behavioral and molecular bases of such divergence were traditionally investigated in human twin studies. However, the difficulty in conducting longitudinal studies in humans, as well as the limited range of phenotypes that could be assessed in each twin cohort, leave many fundamental questions open. In particular, underlying mechanisms at the levels of cells, tissues, systems or the entire brain and their interaction across these scales cannot be determined in human subjects because it is not possible to collect all relevant phenotypes with sufficient depth and precision or to manipulate the processes in question experimentally. Thus, addressing these problems calls for a suitable animal model in which both environment and genotype can be strictly controlled.

Individualization involves an increasingly differing response of initially highly similar individuals to exposure to seemingly the same environment. We propose that activity-dependent structural plasticity is a central mechanism contributing to the individualization of the brain. The iterative nature of the feedback loops between plasticity and behavior result in increasingly different brains, behavioral trajectories and life courses. In this model, small initial differences are augmented through self-reinforcement. In support of this hypothesis, we previously showed that large groups of isogenic mice that were exposed to an enriched environment (ENR) developed stable and unique social and exploratory behavioral patterns that diverged between individuals over time (*Freund et al., 2013*;

**eLife digest** Even identical twins who share genetics and the same environment develop individual traits. But how individuality emerges and the biological mechanisms behind it are not clear. It is hard to study people for a long time, so scientists turn to animal studies to answer such questions. One way to study the respective effects of genes and the environment is to study differences in genetically identical mice housed in either small cages with few animals and little to do, or in larger cages with toys and more animals. Comparing how these different environments affect individual animals and their biology may help scientists understand individuality.

If individual traits emerge in groups of genetically identical animals housed in the same environment it is likely the result of the individual animal's behaviors or unique experiences. It might also be due to chance. Learning more about the biological processes that underlie individuality may help doctors better match therapies to individuals. It may also help scientists design better studies and help them avoid errors caused by individual variations between animals.

Now, Körholz, Zocher, Grzyb et al. show that living in an enriched environment increases mouse individuality in certain brain and behavioral traits. Other biological traits, like metabolism, did not differ much between the animals in the enriched environment. In the experiments, genetically identical mice housed in either normal laboratory conditions or enriched environments underwent a series of behavioral and biological tests. The mice housed in more interesting environments showed greater variability in how they responded to behavioral tests that exposed them to a new object or an open space than their typically housed peers. There were also more differences in the number of newborn brain cells in the mice living in enriched housing.

These findings may have very important implications for researchers, which could help scientists to better understand how individual behaviors or experiences may affect healthy aging and resilience to disease. Many researchers are also trying to improve the wellbeing of laboratory animals by housing them in more interesting environments. More studies using experiments like those conducted by Körholz et al. may help them understand how enriched animal housing may affect their experiments' results.

DOI: https://doi.org/10.7554/eLife.35690.002

*Freund et al., 2015*). What differed between the mice of this cohort was their unique experience of that same environment and their resulting differential behavior. Because this 'non-shared environment' relates to the individual's own experience and actions, the paradigm revealed a dimension that was previously largely hidden in group effects, but which is of greatest interest for studies addressing sources of variance in a system.

We and others had previously described the stimulatory effect of ENR on mean levels of adult hippocampal neurogenesis (*Kempermann et al., 1997*; *Nilsson et al., 1999*; *Tashiro et al., 2007*), the lifelong activity-dependent generation of granule cells in the mammalian dentate gyrus. Furthermore, we showed that longitudinal individual behavioral trajectories correlated with the within-group differences in numbers of new neurons among the enriched mice, underpinning the suitability of adult neurogenesis as a biologically relevant readout of activity-dependent brain plasticity (*Freund et al., 2013*). This previous experiment suggested an increased variance in the numbers of new neurons integrated into the hippocampal circuit of ENR mice as compared to that of mice living in standard laboratory cages, but the effect could not be claimed unequivocally because the control group was small in size when compared to the experimental group. Moreover, because behavioral assessment was based on monitoring animals in the ENR enclosure, the same constructs were not accessible for control mice. Finally, we could not determine the degree to which the effect of ENR on variance (and hence individuality) was specific to adult neurogenesis and exploratory behavior. The experiment to address these questions is presented here. Because ENR has been shown to influence a broad range of body and brain-related parameters in rodents, including metabolic states (*Wei et al., 2015*), volumes of certain brain areas (*Diamond et al., 1985Diamond et al., 1964*; *Diamond et al., 1966*; *Diamond et al., 1985*), and different behavioral aspects (*Clemenson et al., 2015*; *Garthe et al., 2016*), we were particularly interested in testing the ENR effect on the variance of these parameters. If increases in variance were general across all domains, this would suggest a

common, non-specific causality. From a mechanistic perspective, the paradigm would thus be less feasible as a model that could be used to study the emergence of brain individuality. A more specific and selective induction of variance in response to enrichment would indicate that the observed individualization of the brain does not arise as a mere epiphenomenon of broader effects.

To investigate whether long-term environmental enrichment triggers the specific development of inter-individual differences between mice, we performed a cross-sectional study and analyzed differences in variance between groups of mice housed in one large enriched environment or in control cages (CTRL). Both ENR and CTRL groups consisted of 40 female C57BL/6JRj mice that were randomly assigned to their respective housing conditions, where they stayed for 105 days (*Figure 1A*). In addition to the social complexity introduced by the number of animals in the enrichment cage, the complexity of the ENR was increased by the large size and the compartmentalization of the enclosure (*Figure 1B*). A total of 28 morphological, behavioral and metabolic variables were assessed (*Supplementary file 1* and *2*).

## Results

### ENR reduces mean body size, but does not affect its variance

To determine the effects of the ENR on gross body morphology, we monitored the body weight of all animals over the 105 days of the study (*Figure 2A*). At the beginning of the experiment, no differences in weight existed between the two groups, confirming initial similarity between the randomized experimental mice. However, five weeks after the start of the experiment, ENR mice were significantly lighter than mice housed in control cages (CTRL). The difference in body weight remained constant throughout the experiment and indicated, together with the significantly shorter body length in ENR mice (*Figure 2B*), that housing of mice in ENR reduces body size. By contrast, no differences in brain weights were detected between ENR and CTRL mice (*Figure 2C*). The groups did not differ in the variances of body length, body weight and brain weight at any measured time point, suggesting that long-term ENR does not stimulate the development of inter-individual differences in gross body morphology.

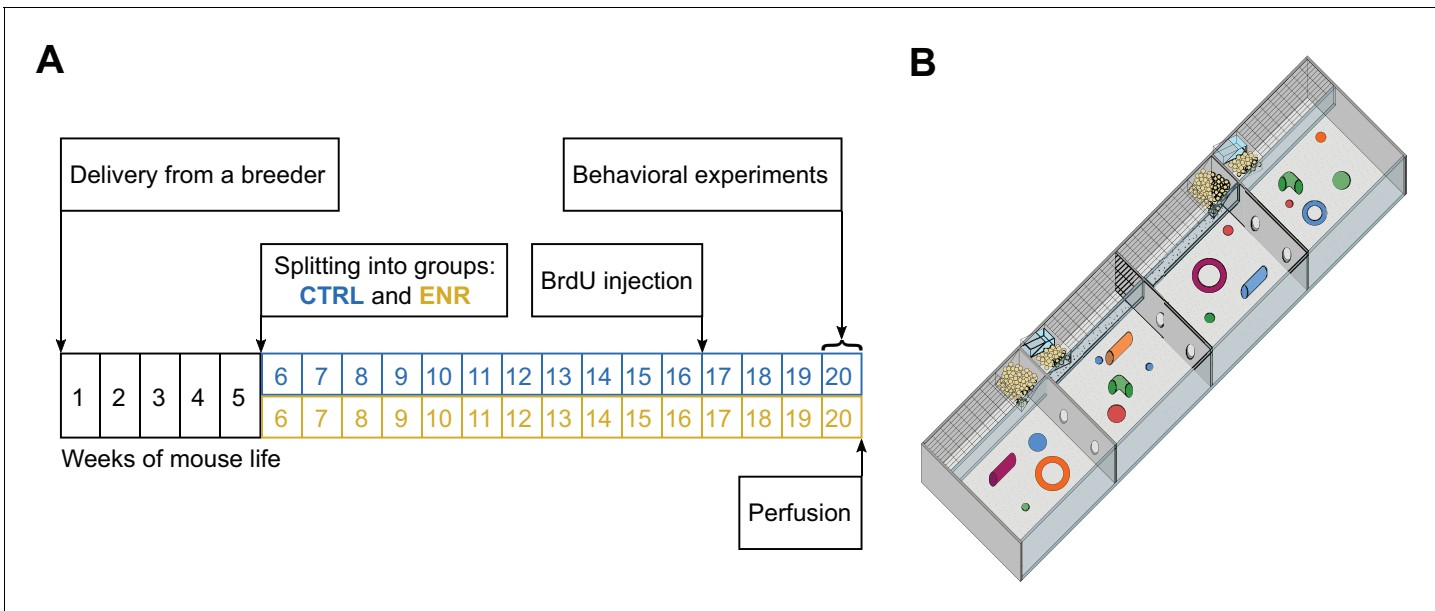

**Figure 1.** Experimental setup. (**A**) Experimental outline. At an age of 5 weeks, 80 female mice were split equally into two groups: one group lived in an enriched environment (ENR) for 15 weeks and one group lived in standard mouse cages in groups of five mice per cage (CTRL) for the same period of time. To analyze neurogenesis in the hippocampus, mice received intraperitoneal BrdU injections three weeks before perfusion. Behavioral phenotyping was performed in the last eight days before perfusion. (**B**) The enriched environment enclosure covered a total area of 2.2 m² and consisted of four sub-compartments, which were connected via tunnels. Food, toys and nesting material were provided in every compartment.
DOI: https://doi.org/10.7554/eLife.35690.003

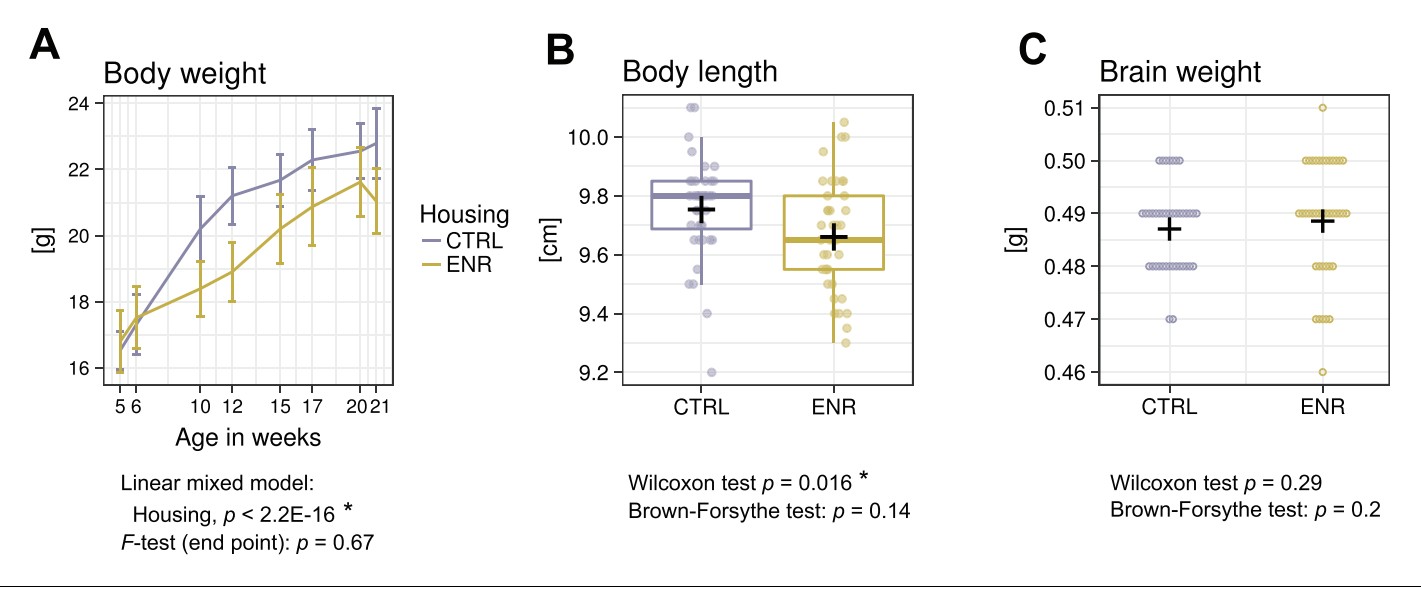

**Figure 2.** Environmental enrichment does not increase variance in gross body morphology. (A) Longitudinal measurement of body weight. Presented are means ± standard deviations. (B) Body length and (C) brain weight were assessed at the end of the experiment. Box and whisker plot: center line, median; plus sign, mean; upper and lower hinges, first and third quartiles; whiskers, highest and lowest values within 1.5 times the interquartile range outside hinges; dots, individual data points. Asterisks indicate significant effects at 5% threshold in the indicated statistical tests. Full information on the statistical tests is available in *Supplementary file 2*. CTRL, control group; ENR, enriched group.
DOI: https://doi.org/10.7554/eLife.35690.004

## ENR increases inter-individual behavioral differences specifically in exploratory behavior

To analyze whether ENR increased inter-individual variability in behavior, all mice were subjected to the open field (OF), novel object recognition (NOR) and rotarod tests (*Figure 3A–B*). ENR mice traveled longer distances during the first OF trial, but showed less locomotion in the second OF trial (*Figure 3C*) and throughout the NOR test (*Figure 3D*). No significant differences were found in the variance of locomotion in any of the OF or NOR trials. In our previous work, we introduced roaming entropy (RE) as a measure of the territorial coverage and exploratory activity of mice in order to introduce a qualitative aspect into activity measurements (*Freund et al., 2013*; *Freund et al., 2015*). To investigate the effects of ENR on spatial exploration, we computed RE for all mice in the OF arena (*Figure 3E–F*). On both days, ENR mice had significantly lower RE than CTRL animals. Moreover, on day 2, ENR mice showed a significantly greater variance in RE, suggesting a higher range of habituation to the OF among ENR mice. Indeed, both ENR and CTRL animals habituated to the OF arena, as indicated by a decrease in RE between the trials (*Figure 3G*). However, habituation was more pronounced and exhibited higher variance in ENR mice. In the NOR test, ENR mice showed a significantly higher variance in the duration of their exploration of the objects when compared to CTRL mice (*Figure 3H*), indicating that ENR increases the inter-individual variability in exploratory behavior. Although some individuals among ENR mice explored objects for much longer than any of the control animals, the median of the entire ENR group was not shifted compared to that of the CTRL group. Finally, to examine the effect of ENR on the recognition memory of individual mice, we analyzed the ability to discriminate a new object from an old one in the NOR test. A trend towards a preference for the new object was found only in the ENR mice and not in the CTRL group (*Figure 3I*).

The performance of ENR mice on the rotarod was superior to that of CTRL mice in all trials (*Figure 3J*), indicating that ENR stimulates motor coordination and, presumably, fitness. Initially, ENR mice also showed a greater variance in their performance compared to that of CTRL, but while both groups improved in the task, this difference gradually disappeared.

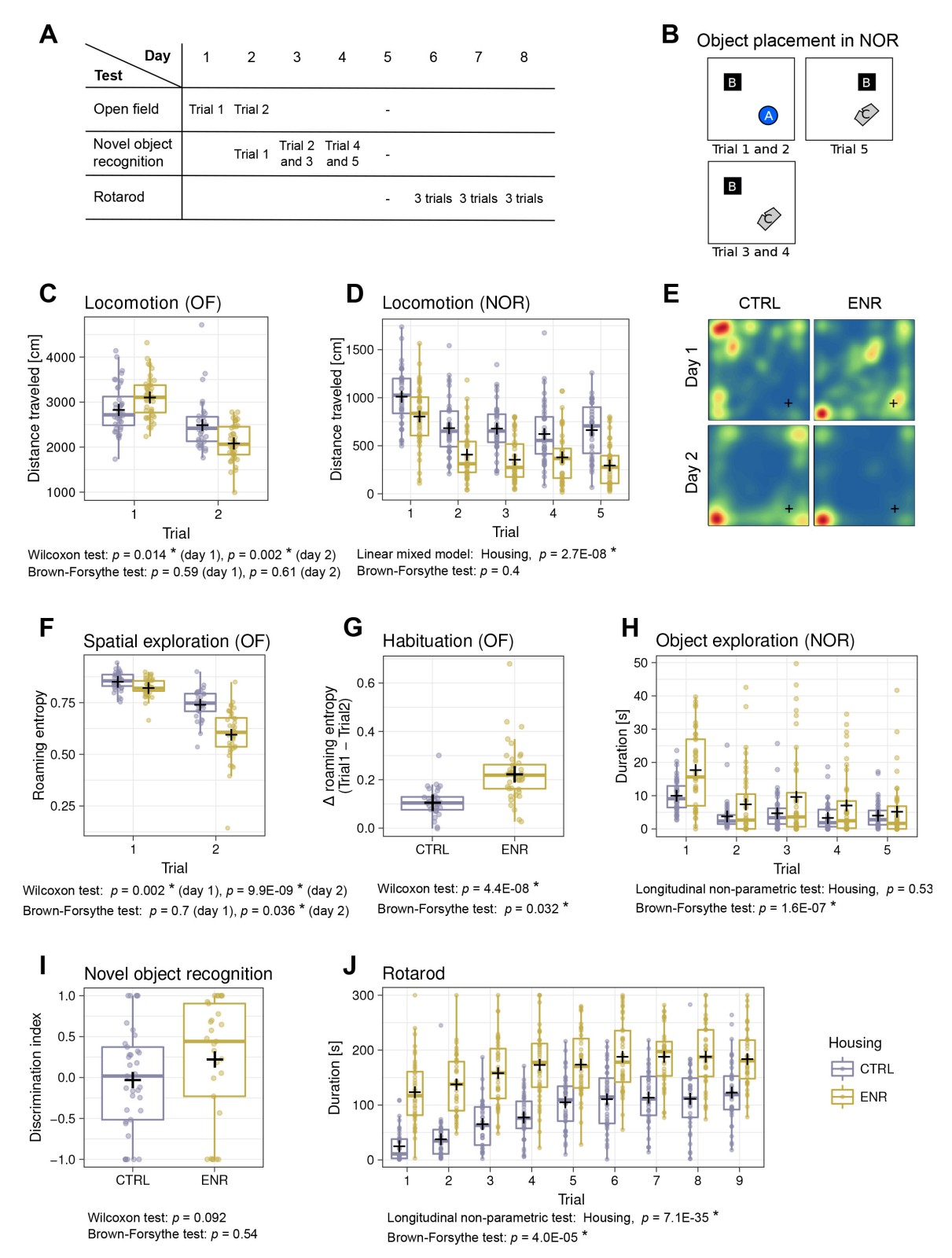

**Figure 3.** Mice living in an enriched environment exhibit inter-individual differences in motor abilities, spatial exploration and object exploration. (**A**) Timeline of behavioral testing. (**B**) Object placement in trials of the novel object recognition (NOR) task. (**C**) Total distance that control (CTRL, blue) and enriched animals (ENR, yellow) moved in the arena on the two days of open field (OF) testing. (**D**) Total distance that mice moved during each trial in the NOR test. (**E**) Representative heatmaps for two mice depicting the probabilities that each mouse was to be found at a specific location in the OF

*Figure 3 continued on next page*

*Figure 3 continued*

arena. Blue indicates lowest and red highest probabilities, respectively. The corner in which the light source was located is marked with a cross (+). (F) Roaming entropy in the OF arena describes spatial exploration. (G) Habituation to the OF expressed as a difference in roaming entropy between trials. (H) Object exploration in the NOR test. (I) Discrimination index indicating preference for the novel (+1) or the old (−1) object. (J) Duration that mice spent on the rotating rod during the individual trials of the rotarod task. Box and whisker plots, see *Figure 2*. Asterisks, significant effects at 5% threshold.

DOI: https://doi.org/10.7554/eLife.35690.005

The following figure supplements are available for figure 3:

**Figure supplement 1.** Photographs showing examples of the OF arena with objects (labeled in yellow) during NOR testing.

DOI: https://doi.org/10.7554/eLife.35690.006

**Figure supplement 2.** Rotarod mean duration from all trials (A) and daily sessions (B).

DOI: https://doi.org/10.7554/eLife.35690.007

Together, we conclude that ENR promotes the development of inter-individual differences in specific interactions with the environment, but not in pure locomotor activity.

## ENR increases inter-individual variability in the survival of new-born neurons

To assess whether the observed behavioral variability is reflected in differences in brain plasticity, we quantified the rates of adult neurogenesis in the dentate gyrus of the hippocampus. To estimate the proliferation of precursor cells, we stained mouse brain sections for the proliferation marker Ki67 (*Figure 4A–B*), whereas new-born cells that survived initial selection processes were identified by the presence of BrdU, which was injected 3 weeks before the end of the experiment (*Figure 4C– E*). No differences in the means or variances of the numbers of proliferating cells in the subgranular zone of the dentate gyrus were observed between ENR and CTRL mice (*Figure 4F*). By contrast, we found a significant increase in the means and variances of the numbers of BrdU-positive cells in animals housed in ENR (*Figure 4G*), highlighting the specific effect of ENR on the survival of new-born cells. Co-localization of BrdU-positive cells with the neuronal marker NeuN and the astrocytic marker S100β (*Figure 4E*) showed that the variances in the survival of both neurons and astrocytes were higher in the ENR group than in the CTRL animals (*Figure 4H–I*). An increase in the total number of cells was, however, only found in the neuronal cell population. These results indicate that ENR increases inter-individual variability in the survival of new-born neurons and astrocytes but not in the proliferation of precursor cells in the dentate gyrus.

## ENR does not elicit increases in variances of the hippocampus and cerebral cortex sizes

ENR has been long known to induce broad changes in brain structure in rodents, such as thickening of the cerebral cortex (*Diamond et al., 1966*) and increases in the volume of the dentate gyrus (*Kempermann et al., 1997*). We also showed that the volume of the mossy fibers increases upon environmental stimulation concomitantly with adult neurogenesis in mice (*Römer et al., 2011*). To further assess whether ENR increases inter-individual variability in brain plasticity beyond adult neurogenesis, we estimated the volumes of the hippocampus and its substructures: the dentate gyrus, infra- and suprapyramidal mossy fiber tracts (IMF and SMF) and the hilus (*Figure 5A–D*). The volume of the entire hippocampus did not differ between ENR and CTRL mice (*Figure 5H*), but the volume of the dentate gyrus was significantly increased in ENR mice (*Figure 5I*). Furthermore, IMF (*Figure 5J*) and the hilus (*Figure 5L*), but not SMF (*Figure 5K*), were significantly larger in ENR animals than in CTRL mice. None of these parameters showed different variances depending on housing conditions. This suggests that ENR differentially influenced various aspects of hippocampal plasticity, while showing that the increased inter-individual variability that was triggered by ENR was specific to adult neurogenesis.

Next, we measured the thickness of the entorhinal, cingulate and motor cortex (*Figure 5E–G*) as enrichment might specifically increase cortex thickness and structure in these areas (*Diamond, 2001Diamond et al., 1964*, *Diamond, 2001*). Although we did not detect differences in the thickness of any of these cortices between CTRL and ENR mice (*Figure 5M–O*), the motor cortex thickness showed a significantly higher variance in the ENR group (*Figure 5O*).

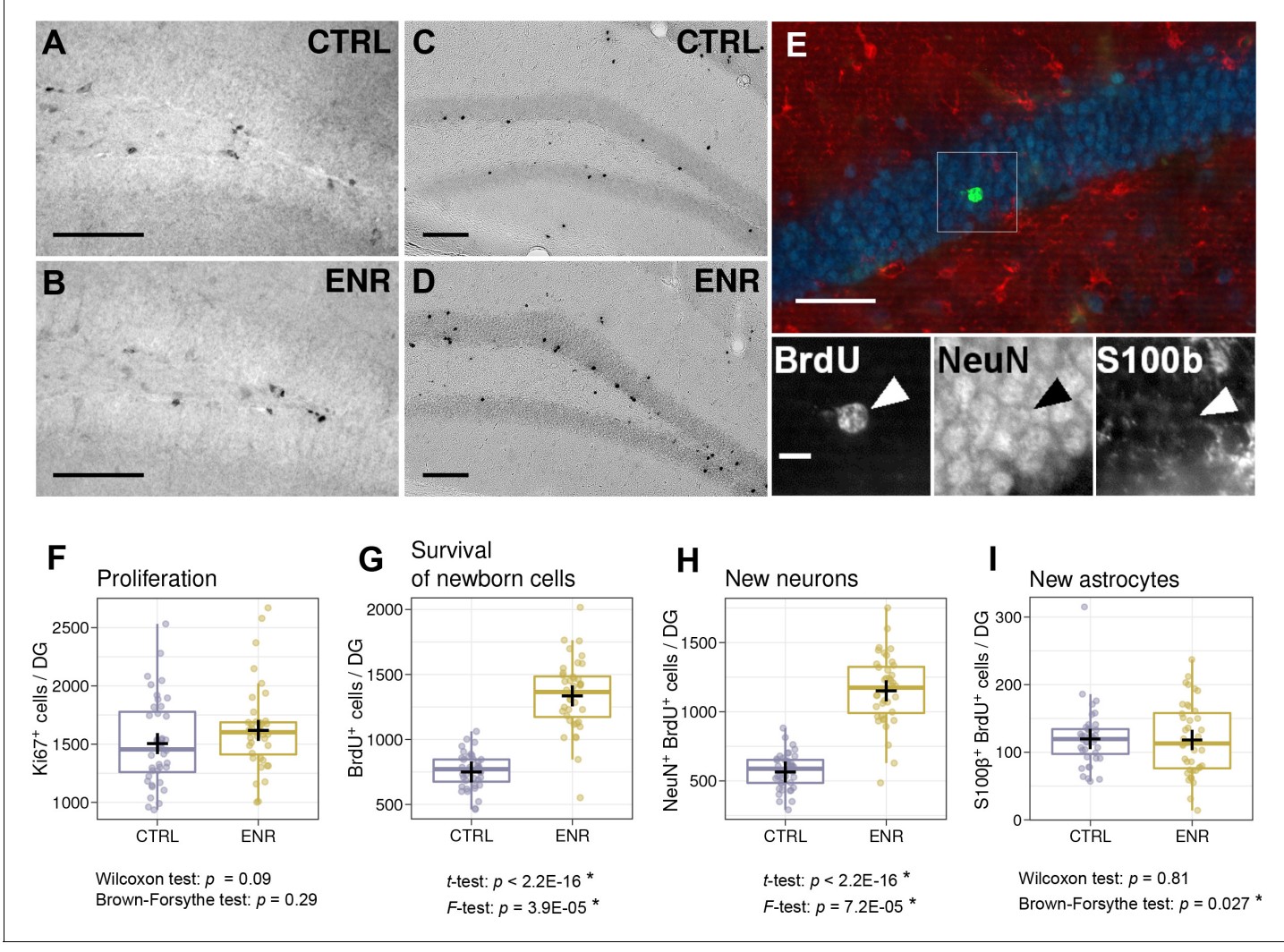

**Figure 4.** Environmental enrichment leads to the development of individual levels of adult hippocampal neurogenesis. (A, B) Representative images of Ki67 immunostaining, which marks proliferating cells in control (CTRL) and enriched (ENR) mice. (C, D) New-born cells were identified by BrdU immunoreactivity three weeks after the injection of BrdU. (E) The proportions of new-born neurons and astrocytes were determined by co-localization of BrdU (green) with NeuN (blue) and S100β (red), respectively. The image shows a single optical section. The arrowhead highlights a new-born neuron. (F) No difference in the number of proliferating cells can be observed between mice housed under CTRL and ENR conditions. (G–I) ENR mice have significantly higher means and variances in the numbers of new-born BrdU-positive cells (G) and new neurons (H), whereas only the variance of the number of new astrocytes was increased (I). Scale bars are as follows: (A–D) 100 μm; (E), 50 μm; (E inset), 10 μm. Box and whisker plots, see *Figure 2*. Asterisks indicate significant effects at a 5% threshold.
DOI: https://doi.org/10.7554/eLife.35690.008

## ENR reduces organ weights and cholesterol levels, but does not induce metabolic variability

Since ENR reportedly had beneficial effects on metabolism in outbred mice (*Wei et al., 2015*), we compared the weights of the liver and adrenal glands as organs playing a role in metabolic and hormonal regulation and analyzed basic blood biochemistry. In agreement with lower body weights, ENR animals had smaller adrenal glands and livers (*Figure 6A–B*). The levels of plasma corticosterone, which is synthesized in the adrenal gland and is used as an indicator of animal stress, did not differ between the groups (*Figure 6C*). Moreover, significantly lower plasma cholesterol levels (*Figure 6D*) in ENR mice but no differences in triglyceride and glucose levels were seen when comparing ENR and CTRL mice (*Figure 6E–F*). No variance differences between the two groups were detected in any of the measured metabolic parameters.

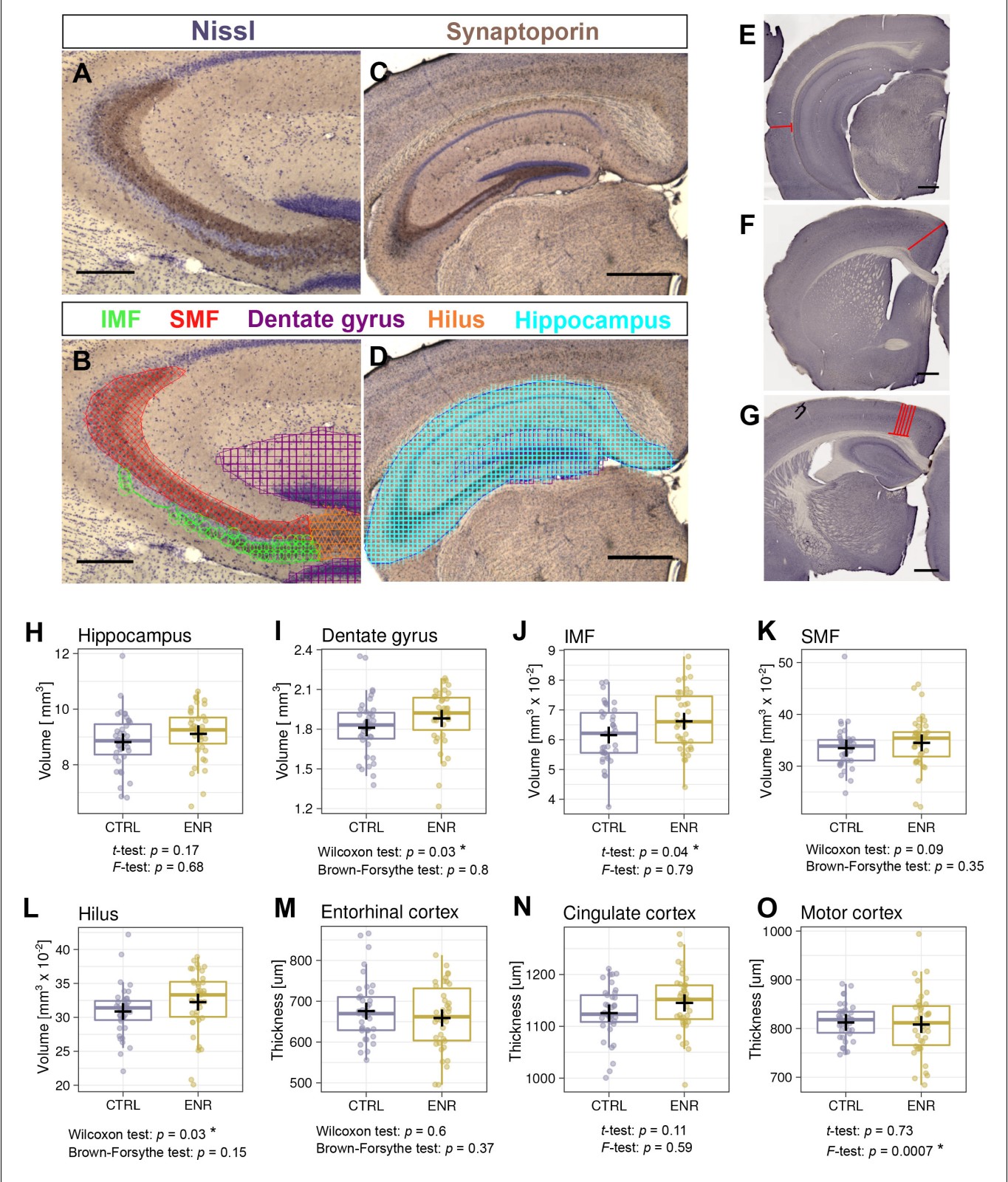

**Figure 5.** Environmental enrichment does not increase variability in gross brain morphology. (**A–D**) Representative images of sections immunostained with a synaptoporin antibody (brown) and counterstained with Nissl (purple). (**B, D**) Examples of contour tracing with overlaid Cavalieri probe estimator markers for the indicated brain structures. (**E–G**) Thickness measurement on Ki67-DAB stained sections of entorhinal (**E**), cingulate (**F**), and motor cortex (**G**). (**H–L**) Results from volumetric analyses of the hippocampus (**H**), dentate gyrus (**I**), infrapyramidal mossy fiber tract (IMF; **J**), the suprapyramidal mossy

*Figure 5 continued on next page*

*Figure 5 continued*

fiber tract (SMF; **K**), and the hilus (**L**). (**M–O**) Thickness of three cortical areas: the entorhinal cortex (**M**), the cingulate cortex (**N**) and the motor cortex (**O**). Scale bars are 200 μm in (**A-B**), and 500 μm in (**C-G**). CTRL, control; ENR, enriched mice. Box and whisker plot, see *Figure 2*. Asterisks indicate significant effects at 5% threshold.

DOI: https://doi.org/10.7554/eLife.35690.009

## ENR restructures relationships between phenotypes

To analyze the impact of ENR on relationships between phenotypes, we calculated correlations separately for CTRL and ENR mice (*Figure 7*). Strong correlation between two traits would point towards shared regulatory mechanisms in the control of such parameters. We observed few significant correlations in either housing condition, which suggests that the majority of the traits were independent of each other. In both animal groups, we found significant positive correlations of roaming

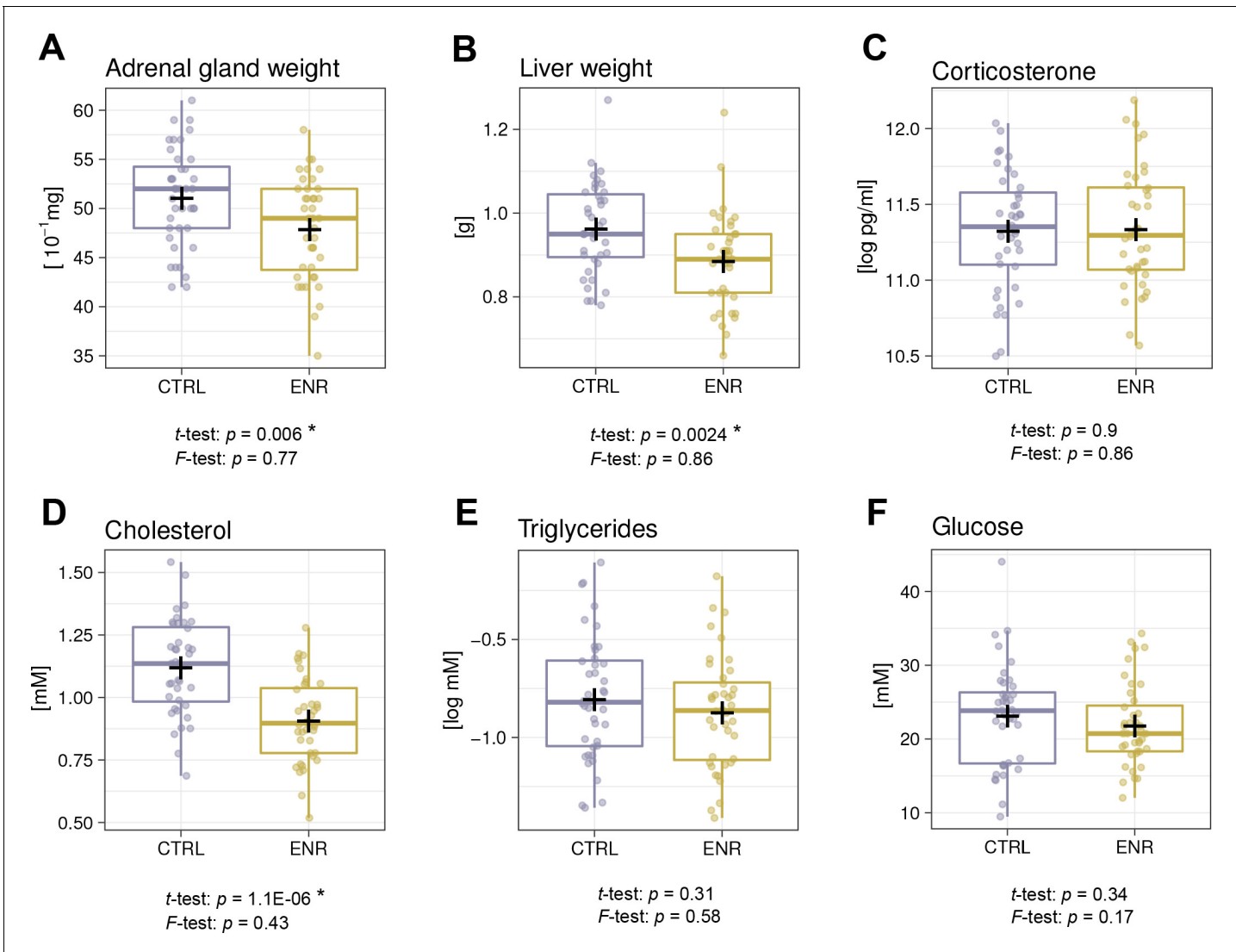

**Figure 6.** Environmental enrichment does not induce metabolic variability. (**A, B**) Environmental enrichment (ENR) mice have adrenal gland (**A**), and liver (**B**) weights that are lower than those of control (CTRL) animals. (**C**) Housing does not affect acute corticosterone levels. (**D–F**) Effects of ENR on plasma biomarkers: cholesterol (**D**), triglycerides (**E**) and glucose (**F**). Box and whisker plots, see *Figure 2*. Asterisks indicate significant effects at a 5% threshold.

DOI: https://doi.org/10.7554/eLife.35690.010

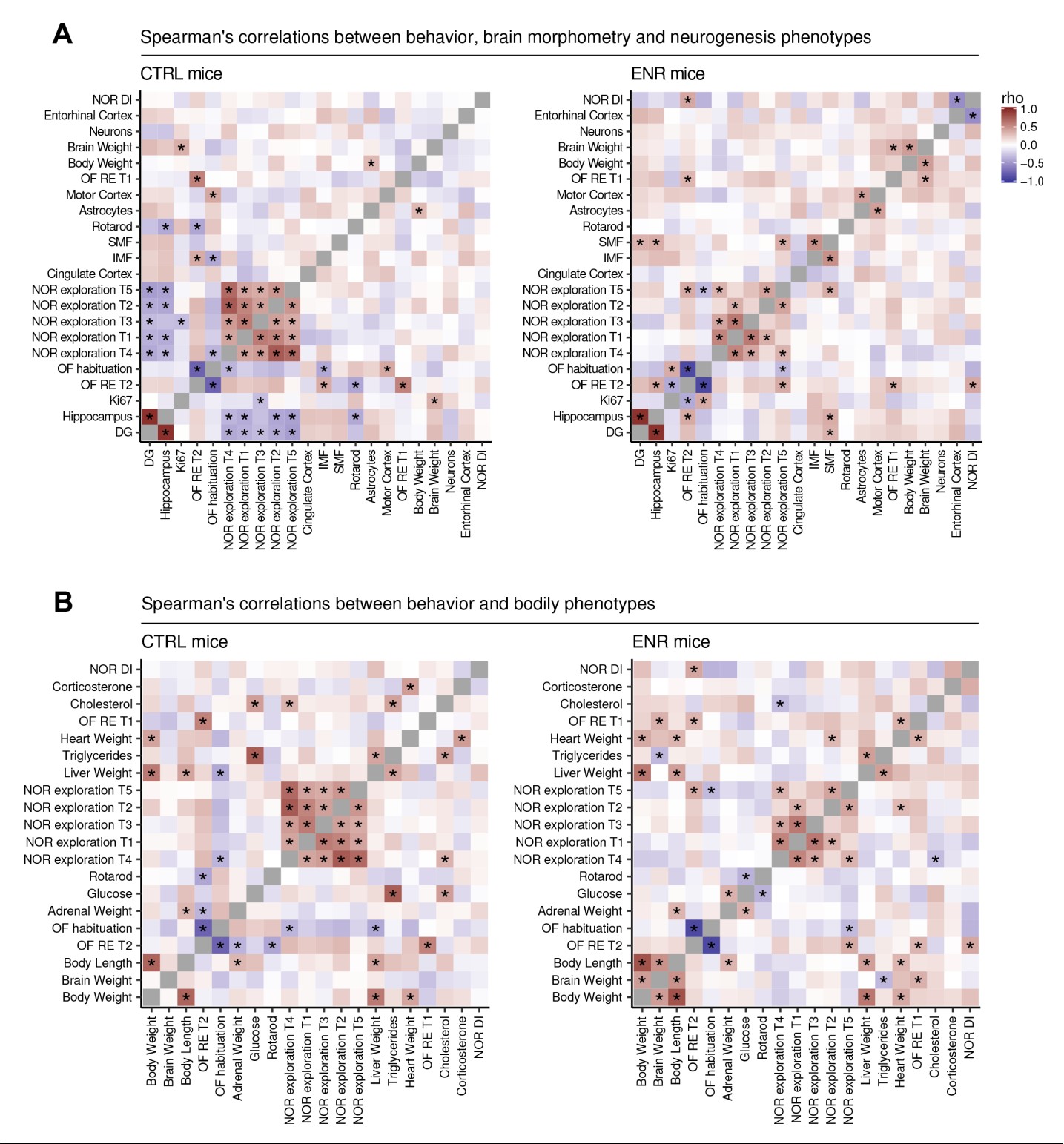

**Figure 7.** Environmental enrichment leads to partial restructuring of relationships between phenotypes. Heat maps show Spearman's rank correlation coefficient between selected behavioral, morphometric and neurogenic (A) or metabolic traits (B) within the control (CTRL, left panels) and the enriched (ENR, right panels) groups. Rotarod performance was summarized as a mean of data from all individual trials for each animal. Phenotypes were ordered on the basis of hierarchical clustering in the ENR group. Significant correlations (p<0.05) are marked with asterisks. NOR, novel object recognition test; OF, open field test; T1–5, trials 1–5 of the NOR and OF tests; RE, roaming entropy; DG, dentate gyrus; DI, discrimination index. Source files listing the rho values are available in *Figure 7—source data 1* (CTRL) and in *Figure 7—source data 2* (ENR).

*Figure 7 continued on next page*

*Figure 7 continued*

DOI: https://doi.org/10.7554/eLife.35690.011

The following source data is available for figure 7:

**Source data 1.** Correlation matrix, CTRL mice.
DOI: https://doi.org/10.7554/eLife.35690.012
**Source data 2.** Correlation matrix, ENR mice.
DOI: https://doi.org/10.7554/eLife.35690.013

entropy and object exploration between trials of the OF and NOR tests, respectively (*Figure 7A*). These correlations indicate that the recorded behaviors were reliable and characteristic for each individual. ENR, however, weakened correlations between trials in NOR, as well as the negative association of object exploration to OF habituation, hinting towards more specific responses of animals to the environment (e.g. exposure to novel objects or their placement). Housing in ENR led to remodeling of the associations between the brain structures and behavior (*Figure 7A*). ENR uncoupled the negative correlations of object exploration in NOR test and of rotarod performance to the volume of the hippocampus. Similarly, habituation to the OF arena was negatively associated with the size of the IMF and positively correlated with the motor cortex thickness in CTRL mice, but not ENR mice. Hippocampal neurogenesis did not show significant correlation with any of the assayed phenotypes.

Metabolic phenotypes, namely plasma glucose, cholesterol and triglycerides were positively correlated in both housing conditions, but these relationships were weakened by ENR (*Figure 7B*). As expected, plasma triglycerides correlated positively to the liver size in both groups. Epidemiological studies in humans suggest that brain and cognition are linked to metabolism (*Kapogiannis and Mattson, 2011*; *Panza et al., 2012*). We observed few associations between the measured phenotypes in our mice (*Figure 7B*). ENR changed the sign of the correlations between object exploration in NOR test and plasma cholesterol from positive to negative. It also promoted negative correlation between plasma glucose and rotarod performance, suggesting that the fitness acquired by ENR mice has a metabolic component.

## Discussion

As medicine acknowledges inter-individual differences as a key determinant in diagnosis and treatment, understanding the biological mechanisms underlying individuality becomes increasingly important. Here we established that ENR is a suitable model to dissect processes leading to brain individualization. The purpose of our multivariate cross-sectional study of ENR effects in mice was to provide an insight into the magnitude of the individualization of phenotypes spanning broad aspects of physiology. We have shown that different traits are not uniformly affected by the stimuli and, besides effects on the mean, effects on variance could also be observed for certain parameters. The effects in which ENR increases differences between individuals in the group cluster in variables related to behavior and adult neurogenesis.

To evaluate the effects of ENR, we used statistical tests that were appropriate for each given data distribution (as described in the 'Materials and Methods') and interpreted *p*-values smaller than 0.05 as sufficient evidence for the influence of ENR on a particular phenotype. *p*-values were introduced by Ronald A. Fisher as an informal index to indicate whether or not the null hypothesis of no effect fails to account for the whole of the observations (summarized in *Lehmann, 1993* and in *Goodman, 1993*). Lower *p*-values imply lower likelihood of the null hypothesis given the obtained data, with strength of evidence being interpreted as weak to moderate in the range between 0.1 to 0.01, and strong to very strong at or below 0.001 (*Goodman, 1999*). The negative findings, though, are less straightforward to evaluate. With 40 individuals in each experimental group, a 5% pre-set significance level in a two-tailed variance test enabled us to detect an effect size of 2.5 (ratio of variances) with power of 0.8. This is a relatively large effect and moderate power, which implies that smaller changes in variance, as well as some of the stronger effects, could have been missed. At the same time, we would argue that the identified effects are robust enough to be biologically relevant. Furthermore, we adopted marginal, that is individual, interpretation of responses to ENR for each of

the assayed phenotypes and did not control for an experiment-wide type I error rate. Correction of *p*-values upon multiplicity of tests stems from posing the universal null hypothesis that no association exists between any pair of variables under investigation (*Rothman, 1990*). The question we posed was, however, not whether ENR influences the variance or mean of *any* trait (a universal null hypothesis), in which case the control of experimental error rate would be necessary, but rather each of the specific responses was of interest (for the distinction of various scenarios see *Cook and Farewell, 1996*). Most importantly, however, the correction for multiple tests leads to inflation of type II errors (false negatives) and hence introduces a 'penalty for peeking', that is, the more parameters are investigated, the less likely each of the true associations is to be detected (*Perneger, 1998*; *Rothman, 1990*). Significant *p*-values in our experiment were not uniformly distributed across all phenotypes, supporting our view that we achieved a balance between false-negative and false-positive error rates sufficient to provide a coherent overview of the effects of ENR on a wide variety of traits.

Our behavioral data highlight a significant effect of ENR on animals' active interaction with their environment: improved fitness and coordination, as assessed by the rotarod task, and modified patterns of exploratory activity and habituation in OF and NOR tests. It has long been known that ENR elicits profound effects on brain plasticity and behavior (*Mohammed et al., 2002*; *Sale et al., 2014*). The effects of ENR housing on animal behavior in variations of OF and NOR tests have been reported in earlier studies. We have previously found, for example, that ENR mice habituate faster to an open field, and this observation has been interpreted as improved spatial processing (*Kempermann and Gage, 1999*). Since Ennaceur and Delacour introduced the spontaneous object recognition task (*Ennaceur and Delacour, 1988*), its modifications have been used to dissect components and neural bases of recognition memory (reviewed in *Ameen-Ali et al., 2015* and *Antunes and Biala, 2012*). Short 2 min trials in our NOR task precluded efficient familiarization with objects (*Melani et al., 2017*), which explains the lack of preference towards the new object. The multi-trial paradigms were developed to reduce extra-experimental variance and thus to improve reproducibility and reduce the numbers of animals needed for the experiments (*Albasser et al., 2010*; *Ameen-Ali et al., 2012*). Although not a multi-trial paradigm in this classical sense, our design involved multiple trials of the NOR and OF tests, which allowed us to confirm that the exploratory behaviors were stable and idiosyncratic, as indicated by the high intra-group correlations between the trials of each task. The current study now highlights that ENR also induced substantial inter-individual variability in specific behavioral parameters. Particularly in the NOR test, ENR mice showed much greater variability in object exploration times compared to the relatively homogenous CTRL group. In the OF test, ENR mice not only exhibited greater habituation to the arena, but this response was also more variable than that in CTRL animals. Locomotion, a less specific aspect of exploration, decreased in ENR mice and its variance was not affected by housing. These observations corroborate our previous finding of ENR-induced individualization of spontaneous interactions of mice with their environment (*Freund et al., 2013*). We have previously argued that the active participation with the outer world and habitat that manifests itself in the individual range of locomotion within that world (roaming entropy) is a major driving force of brain plasticity, presumably not limited to adult hippocampal neurogenesis (*Freund et al., 2013*). The current data are in agreement with this hypothesis. The improvement of rotarod performance in the ENR mice observed in this study implies that, even if the running wheels are not supplied, large cage area and toys to interact with provide considerable motor stimulation (*Kempermann and Gage, 1999*), which, together with elevated variance of the motor cortex thickness, suggest that ENR strongly affects the plasticity of motor responses. It has been proposed that activation of motor cortex by ENR has widespread modulatory effects on other cortical areas (*Sale et al., 2014*; *Di Garbo et al., 2011*; *Niell and Stryker, 2010*).

Our previous report suggested that long-term ENR induces variability in the survival of new-born neurons (*Freund et al., 2013*). The large size of both the control and the ENR groups in the present study allowed us to corroborate that finding (*F*-test: p=0.00004). Although new astrocytes did not increase in numbers, we observed a small effect on the variance of these. By contrast, there were no differences in the variance of the number of proliferating cells between ENR and CTRL animals. Although the effect of ENR on the mean numbers of proliferating cells did not reach the significance threshold applied here (p=0.09), together with our previous report (*Kempermann et al., 2002*), this study suggests that over prolonged periods of time, enrichment might have subtle pro-proliferative effects. Proliferating cells are a substrate on which selection mechanisms can act, but their behavioral

significance as such has not yet been shown. The fact that ENR does not trigger individuality in pre-cursor cell proliferation indicates that individualization mechanisms act selectively only on those aspects of neurogenesis that are relevant for interactions with the environment. In contrast to the study by *Freund et al., 2013*, in which 21% of the variance in adult neurogenesis could be explained by differences in cumulative roaming entropy (RE), an aggregated measure of the longitudinal behavioral trajectory, the current analysis detected no correlation between the number of newborn neurons and any of the cross-sectionally measured behaviors, including RE measured in the OF test. RE is a convenient single parameter describing the uniformity of coverage of a given space (*Freund et al., 2013*; *Freund et al., 2015*), but its interpretation depends on the context in which it is used. Here, we calculated RE to describe the exploration in the OF, because it carries more infor-mation than the traditional measures. Specifically, it does not rely on hard boundaries, such as the periphery and center of the open field, and also takes into account exploration within each of these zones. (For example, a mouse that remains in the corner of an OF arena and a mouse that visits the entire perimeter of an arena might have spent the same amount of time in the periphery but differ in the RE measure.) In the present study, RE in the OF was lower among ENR mice than in CTRL mice. The behavior in the familiar environment of the ENR cage, however, is presumably driven by other factors and can be influenced by the group interactions (*Shemesh et al., 2013*), whereas the OF test reflects an individual response of an animal to novel situations and might, therefore, con-stitute a different construct. Perception of novelty and incentive to explore an empty OF arena are likely to be different for CTRL animals, which spent their entire life in small cages, and the ENR group.

The mossy fiber projection, and especially its infra-pyramidal blade (IMF), is highly plastic in mice (*Crusio et al., 1989*; *Schwegler et al., 1981*) and ENR can modulate its size (*Römer et al., 2011*). We found an increased volume of the IMF but no effect on the variance of this volume after 3 months of ENR, suggesting that even for aspects of hippocampal plasticity there is no simple paral-lelism in the effects of ENR. Similarly, we observed an increase in the volume of the dentate gyrus, but did not detect changes in the mean volume of the hippocampus or in the variance of this pheno-type. Adult-generated neurons contribute to the IMF (*Römer et al., 2011*), yet we did not observe a correlation between numbers of new neurons and IMF volume within either housing group, impliying that, under physiological conditions, mechanisms other than adult neurogenesis determine the bulk of the IMF. This finding is in agreement with the results from the screen in the mouse genetic reference population (*Krebs et al., 2011*).

Cortex thickness changes upon enrichment of the environment have been reported in the older literature and were the cornerstone of the growing impact of the ENR paradigm in the 1970s (reviewed in *Diamond, 2001*; *Rosenzweig and Bennett, 1996*). Increases in cortical thickness do not strictly mirror volume changes (*Hammelrath et al., 2016*; *Winkler et al., 2010*), but they are an indication of massive cellular rearrangements in the cerebral cortex (*Diamond et al., 1964Diamond et al., 1966*). In our study, the only effect that we found in response to ENR was an increase in the variance of the motor cortex thickness. The key difference between our work and classical studies is that we worked with mice, whereas essentially all classical studies had been done in rats. The dynamics of three-dimensional brain development during first months of life differ between these two species (*Hammelrath et al., 2016*). Furthermore, the majority of old experiments compared enriched animals to impoverished littermates, which were kept in social isolation. Such impoverishment negatively affects brain size (*Fabricius et al., 2010*), thus amplifying the relative effects of enrichment (*Bennett et al., 1964*). We believe that the impact of ENR on cortical plasticity deserves still more specific analyses with much greater resolution.

It had been shown that ENR influences metabolism (*Wei et al., 2015*): keeping outbred mice in ENR resulted in decreased body weight, mostly through reducing fat content; lowered blood choles-terol, triglycerides, and glucose; and improved insulin and leptin signaling. It has to be noted that the cages in the experiments performed by Wei and colleagues were equipped with running wheels to stimulate physical exercise. In the present study, we also observed decreases in body and liver weights, as well as lowering of plasma cholesterol, which indicates that ENR alone has a moderate beneficial effect on metabolism even in the absence of intense physical exercise. Although we rou-tinely recorded reduced body weights in mice living in ENR conditions (*Kempermann and Gage, 1999* and unpublished observations), this response might be subjected to local conditions that are unique to specific animal facilities (*Crabbe et al., 1999*) as no such effect was observed in our

previous experiment (*Freund et al., 2013*; *Freund et al., 2015*). ENR animals also had smaller adrenal glands and even though corticosterone levels were similar, this points towards reduced stress in ENR mice compared to CTRL. Finally, we did not detect differences in the variances of any of the metabolic parameters, further substantiating the conclusion that individualization of behavior and brain plasticity by ENR is not an epiphenomenon of more global physiological divergence.

Although the issue of variance was brought up in very early studies (*Walsh and Cummins, 1979*), the ENR literature has not been much concerned with variance effects and inter-individual differences. The focus has always been on mean group effects. The question of ENR effects on variance came up, however, in the context of a movement in animal husbandry to provide larger space and enriching cage accessories in order to improve animal wellbeing and to provide more species-appropriate conditions. Variability induced by ENR, the concern went, would work against the desired standardization and stability of animal experiments in the life sciences. A widely cited study in mice by Wolfer et al., however, confirmed that ENR 'increases neither individual variability in behavioral tests nor the risk of obtaining conflicting data in replicate studies' (*Wolfer et al., 2004*). The results presented here (*Figure 3*) stand in clear contrast to the first part of this statement and potentially also the second. As we did not test a full spectrum of behavioral tasks, we must not generalize our conclusion beyond open field and novel object recognition tests (this study), or free roaming in the cage (*Freund et al., 2013*). We would hypothesize that behavioral traits related to exploration and adjusting to novel situations, including hippocampal learning, are more strongly affected than other traits. The conclusion from Wolfer et al. requires a careful qualification. Nevertheless, we fully agree with the overall conclusion that the 'housing conditions of laboratory mice can be markedly improved without affecting the standardization of results', especially if group sizes are sufficiently large. For most variables, even 3 months of ENR did not increase variability or alter correlations with other phenotypes. Furthermore, systemic variation might actually improve reproducibility (*Richter et al., 2011*; see also *Richter et al., 2010*; with comments and re-analysis in *Jonker et al., 2013* and in *Wolfinger and Reanalysis of Richter, 2013*). And finally, the attempt to ignore the within-group variation as an expression of a differential response to the same nominal stimulus might actually contribute to the 'reproducibility crisis' to a much larger extent than previously appreciated.

The mechanisms by which ENR increases variance are currently unknown. We hypothesize that increases in variability are a result of the progressive amplification of initially small inter-individual differences that existed before the start of the experiment, or that were introduced by stochastic events in the initial period of ENR housing. Potential sources of pre-existing variation include prenatal influences on the pregnant mother, intrauterine positioning of the fetus and early postnatal experiences (*Lathe, 2004*). Early life-experiences especially are known to change epigenetic modifications in the brain, which contribute to the long-term control of gene expression. For instance, differences in maternal care in rats led to differences in the DNA methylation state of the glucocorticoid receptor in the hippocampus of the offspring (*Weaver et al., 2004*). Moreover, human twin studies have suggested that monozygotic twins increasingly differ in epigenetic marks from early life to adulthood, presumably as a result of their different experiences (*Cheung et al., 2018*; *Fraga et al., 2005*). Environmental enrichment builds on the initial variation and amplifies the differences by providing opportunity for the development of individual behavior.

According to our hypothesis of positive feedback through experience, the individualization that occurs in the first months of ENR housing should lead to a permanent discordance of behavior, responses to cognitive challenges, and possibly also brain morphology. In support of this, we have previously shown that mice establish stable behavioral trajectories in the first two months of ENR housing that are maintained for the time of monitoring (*Freund et al., 2013*) and are presumably kept up long-term. Whether life-long ENR housing would increase the inter-individual variability even more over time and whether the inter-individual differences are stable after withdrawal of the ENR stimulus are the subjects of current investigations. The positive influence of ENR on neurogenesis and behavior is independent of age (*Kempermann et al., 1998*), but the housing of mice in ENR for longer than 3 months does not further increase experience-dependent neurogenesis (*Kempermann and Gage, 1999*). Concurrently, the exploratory activity of mice was shown to decrease with time in ENR (*Freund et al., 2013*), suggesting, together with the age-related decline in neurogenesis, that the strength of the iterative feedback between neurogenesis and behavior decreases, which could result in a plateauing of the individuality effect with time.

The question arises of whether ENR is unique among activity-dependent plasticity experiences in inducing behavioral and structural divergence. Published studies tend not to provide sufficient information about the individualizing effects of other manipulations because of modest group sizes. ENR is a complex paradigm, in which inanimate aspects of the environment and social interactions intertwine over prolonged periods of time. We hold the view that both this complexity and duration are essential elements in the consolidation of the induced changes. Accordingly, in our previous study (*Freund et al., 2013*; *Freund et al., 2015*), the patterns of general activity, RE and both social and non-social behaviors recorded towards the end of the ENR exposure could not be predicted by the initial differences between animals.

In this study, ENR increased variation within a group of female mice. We have used females to avoid the inter-animal conflict behavior that unrelated males show when put together at the delivery age of 4 weeks and to build on our previous ENR experiments (*Kempermann and Gage, 1999*; *Freund et al., 2013*; *Freund et al., 2015*). Male mice are known to respond similarly to ENR with increased hippocampal neurogenesis (*Zhang et al., 2018*). By contrast, several studies reported sex differences in behavioral responses towards ENR, with female mice being more susceptible to the positive effects of ENR on cognition (*Coutellier and Würbel, 2009*; *Hendershott et al., 2016*; *Wood et al., 2010*). As male mice build stronger hierarchies than females, we expect that the contribution of the social interaction on individuality development is stronger in a male group than in a female group of mice. Dominant males influence the behavior and stress levels of subordinate males (*Curley, 2016*), which could lead to an even stronger and faster individualization in ENR that is less instructed by activity-dependent brain plasticity. On the other hand, increased social distress in subordinate animals might blur individualization effects. However, whether ENR housing leads to the development of inter-individual differences in behavior and brain plasticity in male mice is currently unknown and should be addressed in future experiments.

Despite the ample literature on ENR, few studies addressing larger numbers of dependent variables have been conducted, and to our knowledge, we are first to investigate the interactions between an extended panel of variables in a correlation matrix. Similarly, there has been little insight into the isometry or allometry of the induced changes. Because our experiment employed large groups of animals, we could survey the inter-individual correlation patterns between the variables separately within each environmental condition, thus avoiding spurious relationships that could arise from mean differences between groups. Correlation matrices revealed the extensive relative independence of outcome measures, suggesting that the choice of traits for the analysis was broad enough to reflect distinct underlying causalities. Furthermore, ENR restructured correlation patterns by strengthening or weakening some associations (for details see *Figure 7*), further demonstrating the uneven regulatory influence of ENR on various aspects of physiology and plasticity. Thus, even in the absence of global mean effects on these parameters, ENR seemed to induce broad adaptations in brain plasticity and metabolism.

In conclusion, ENR does not generally increase variability across all domains. ENR-induced increases in variance were specific to exploratory behavior, adult neurogenesis and motor cortex thickness. The correlation pattern of these parameters with other traits was complex, with ENR remodeling many of the associations. We do not think, however, that increased structural variability is limited only to neurogenesis and motor cortex, but rather that the induced changes are very specific and can be revealed only when appropriate aspects of plasticity are examined. ENR arises from this study as a more holistic paradigm than often assumed, and proves to be a decent tool with which to investigate the bases of experience-dependent brain individualization. In the laboratory setting, animals are relieved from pressures present in nature and therefore they are free to choose the degree of interaction with their environment. In our previous longitudinal study, we made the case that in a situation in which both genes and (nominal) environment are kept constant, individuality emerges as a consequence of the so-called 'non-shared environment', that is the individual response to that environment and activity (*Turkheimer, 2011*). This situation is comparable with monozygotic twins, which—even when raised in the same household—develop differences in behavior, appearance and disease susceptibility over time. The underlying mechanisms that drive this divergence are, however, unknown and difficult to address in human studies. Here, we present an animal model that can be used to study the influence of the non-shared environment on individualization and its relation to brain plasticity. Our data suggest that multivariate studies with a large number of individuals and, ideally, a longitudinal design are needed to elucidate the exact

contribution of the non-shared environment to the overall outcome of increased individualization. In perspective, the model of long-term ENR can be extended to analyze the development of individuality in a genetically variable population to provide insights into the interaction of genes with the non-shared environment.

# Materials and methods

**Key resources table**

| Reagent type (species) or resource | Designation | Source or reference | Identifiers | Additional information |
|---|---|---|---|---|
| Strain, strain background (*M. musculus*) | C57BL/6JRj | Janvier Labs | | |
| Antibody | Anti-Ki67 (rabbit polyclonal) | Novocastra | Novocastra: NCL-Ki67p; RRID:AB_442102 | (1:500) |
| Antibody | Anti-BrdU (rat monoclonal) | AbD Serotec | AbD Serotec: OBT0030; RRID:AB_609568 | (1:500) |
| Antibody | Anti-synaptoporin (rabbit polyclonal) | Synaptic Systems | Synaptic Systems:102002; RRID:AB_887841 | (1:500) |
| Antibody | Anti-NeuN (mouse monoclonal) | Merck Millipore | Merck: MAB377; RRID:AB_2298772 | (1:100) |
| Antibody | Anti-S100beta (rabbit monoclonal) | Abcam | Abcam: ab52642; RRID:AB_882426 | (1:200) |
| Antibody | Biotin-conjugated secondary (donkey polyclonal) | Jackson ImmunoResearch | | (1:500) |
| Antibody | Alexa 488-, Cy5-, Cy3- secondaries (donkey polyclonal) | Jackson ImmunoResearch | | (1:500) |
| Commercial assay or kit | Vectastain ABC Elite kit | Vector Laboratories | Vector: PK-6100; RRID:AB_2336819 | |
| Commercial assay or kit | Amplex Red Glucose/ Glucose Oxidase Assay | Invitrogen | Invitrogen: A22189 | |
| Commercial assay or kit | Amplex Red Cholesterol Assay | Invitrogen | Invitrogen: A12216 | |
| Commercial assay or kit | Triglyceride Assay | Abcam | Abcam: ab65336 | |
| Commercial assay or kit | Corticosterone ELISA kit | Enzo | Enzo: ADI-901–097 | |
| Chemical compound, drug | 5-Bromo-2'-deoxyuridine | Sigma Aldrich | Sigma: B5002 | |
| Software, algorithm | Stereoinvestigator 7 software | MBF Bioscience | | |
| Software, algorithm | Ethovision | Noldus | | |
| Software, algorithm | ZEN blue edition | Zeiss | | |

## Animal husbandry

80 female C57BL/6JRj mice were purchased from Janvier at the age of 4 weeks and housed in standard polycarbonate cages (Type III, Tecniplast) in groups of five until the start of the experiment (*Figure 1A*). At the age of 5 weeks, 40 mice were randomly selected and transferred into the enriched environment, where they stayed for three months (no restricted randomization). The number of animals used in each group was decided on the basis of the sample size used in the initial study conducted by *Freund et al., 2013*, which the present study builds on. The enriched environment consisted of four quadratic polycarbonate cages (0.74 × 0.74 m) that were assembled in a row and connected by two plastic tubes each. In total, the enriched environment covered an area of 2.19 m$^2$ (*Figure 1B*). Food and water were provided in every compartment of the cage. To provide sensory stimulation, each compartment of the cage was equipped with plastic toys, tunnels and hideouts, which were cleaned and rearranged once each week. The bedding material was replaced on a

weekly basis. Once a month, the entire enclosure was cleaned. Control animals were housed for the same period of time in standard polycarbonate cages (36.5 × 20.7 × 14 cm) connected to an individually ventilated cage system in groups of five. Control and enriched animals were receiving the same fortified chow (#V1534; Sniff, Germany) with 9% of energy from fat, 24% from protein and 67% from carbohydrates. All mice were maintained on a 12 hr light/12 hr dark cycle with humidity maintained at 55 ± 10% and food and water provided freely. The room was furnished with metal shelves containing laboratory equipment. Three weeks before sacrifice, the mice were injected intraperitoneally with bromodeoxyuridine (BrdU; 50 mg/kg body weight; dissolved in 0.9% NaCl). Injections were performed once per day for three consecutive days. All experiments were conducted in accordance with the applicable European and national regulations (Tierschutzgesetz) and were approved by the responsible authority (Landesdirektion Sachsen).

## Behavioral tests

Before starting the behavioral experiments, every mouse was visibly marked at the tail. To simplify handling, during the morning of every test session enriched animals were placed into standard cages in groups of five, which remained consistent throughout testing, and returned into the enriched environment cage in the evening. Mice were tested in the same order in all behavioral tasks. The sequence of the behavioral experiments is shown in *Figure 3A*.

## Rotarod

Mice were assessed for locomotor abilities using an Economex Rotarod from Columbus Instruments. The rotating cylinder started with a speed of 4 rpm and accelerated by 0.1 rpm. At a final speed of 34 rpm and a maximum time of five minutes, the test was stopped manually. The trial was completed when an animal fell off or reached the maximum duration. The mice were trained on three consecutive days with three trials per day. The rotarod was cleaned after every session.

## Open field test

The open field (OF) enclosure consisted of a 120 × 120 cm square apparatus subdivided into four identical arenas of 60 × 60 cm, allowing for the simultaneous testing of four mice in the apparatus. The 40 cm high white plywood walls were marked with a green tape on the intersections to provide additional spatial clues. The only light source in the room, a 100 watts light bulb, was installed 1.5 m above the intersection of the middle walls, next to the camera (Logitech). Paths were recorded using EthoVision software (Noldus). Mice were placed in the middle of the empty arena and were allowed to explore the arena freely for 5 min in each trial. A total of two trials were performed on two consecutive days. Roaming entropy (RE), a measure of territorial coverage, was calculated according to *Freund et al., 2013Freund et al., 2013*. Each arena was divided into 10 × 10 subfields. The probability $p_i$ of a mouse being in a subfield $i$ was estimated as a proportion of trial time spent in that subfield. Shannon entropy of the roaming distribution was then calculated as:

$$RE = -\sum_{i=1}^{k}(p_i \log p_i)/\log k$$

where $k$ is the number of subfields in the arena ($k = 100$). Dividing the entropy by the factor $\log(k)$ scales the RE to the range from zero to one. RE is minimal for the mice that stay in one place and maximal for the mice that spent equal amount of time in each subfield of the arena. Data from eight CTRL animals were lost in the second trial.

## Novel object recognition test

The two OF trials were considered to serve as habituation for the NOR task (*Figure 3A*). For this task, the same arenas were equipped with two of three different objects: object A was a 1.5 cm high blue cylinder with a diameter of 3.5 cm, object B was a black box of 8.5 × 9.5 × 2 cm, and object C was 4.5 cm long and transparent with a more complex geometric shape (*Figure 3—figure supplement 1*). All objects were made of plastic. For object placement in subsequent trials, see *Figure 3B*. On day 1, following the OF trial, mice were presented with objects A and B. On day 2, the animals were first exposed to the same objects and then in the following trial object A was replaced with object C. The same combination of objects was presented on day 3, followed by a trial in which

object B was moved into the adjacent quarter of the arena. Each trial lasted 2 min. Discrimination index was calculated for trial 3 on the basis of the exploration time for the new and old object as follows: DI = (new object – old object)/(new object + old object), and ranged from −1 (preference for the old object) to 1 (preference for the new object), while 0 indicated no preference (*Miyauchi et al., 2016*). Eleven ENR and three CTRL mice that did not explore object A in any of the first two trials or that did not explore any object in trial 3 were excluded from the calculation of DI.

## Tissue preparation

Two days after the last behavioral experiment was performed, the mice were deeply anesthetized with a mixture of ketamine and xylazine and transcardially perfused with 0.9% NaCl. Directly after the perfusion, the liver, heart and adrenal glands were harvested and weighed. Brains were removed from the skull and postfixed in 4% paraformaldehyde overnight at 4°C and equilibrated with 30% sucrose in phosphate buffered saline (PBS). For immunohistochemistry, brains were cut into 40 μm coronal sections using a sliding microtome (Leica, SM2000R) and stored at 4°C in cryoprotectant solution (25% ethyleneglycol, 25% glycerol in 0.1 M phosphate buffer, pH 7.4).

## Immunohistochemistry

For detection of BrdU-, Ki67- and synaptoporin-positive cells, immunohistochemistry was performed using the peroxidase method as previously described (*Steiner et al., 2008*). Briefly, free-floating sections were incubated in 0.6% $H_2O_2$ for 30 min to inhibit endogenous peroxidase activity. After washing, non-specific antibody-binding sites were blocked using 10% donkey serum and 0.2% Triton-X100 in Tris buffered saline (TBS) for 1 hr at room temperature. For BrdU detection, prior to blocking, sections were incubated in pre-warmed 2.5 M HCl for 30 min at 37°C, followed by extensive washes. Primary antibodies were applied overnight at 4°C as follows: monoclonal rat anti-BrdU (1:500, Serotec), rabbit anti-Ki67 (Novocastra, 1:500), and rabbit anti-Synaptoporin (Synaptic Systems, 1:500). Sections were incubated with biotinylated secondary antibodies for 2 hr at room temperature (1:500, Dianova). Primary and secondary antibodies were diluted in TBS supplemented with 3% donkey serum and 0.2% Triton-X100. Detection was performed using the Vectastain ABC-Elite reagent (9 μg/ml of each component, Vector Laboratories, LINARIS) with diaminobenzidine (0.075 mg/ml; Sigma) and 0.04% nickel chloride as a chromogen. All washing steps were performed in TBS. BrdU- and Ki67-stained sections were mounted onto glass slides, cleared with Neo-Clear (Millipore) and cover-slipped using Neo-Mount (Millipore). BrdU- and Ki67-positive cells were counted, by applying the simplified version of the optical fractionator principle as previously described (*Kempermann et al., 1997*) on every sixth section along the entire rostro-caudal axis of the dentate gyrus, using a brightfield microscope (Leica DM 750). Synaptoporin-stained sections underwent a Nissl-staining before mounting them with Entellan (Merck). To prepare sections for Nissl staining, they were incubated for 20 min in each of the following solutions: staining buffer (4% sodium acetate, 0.96% acetic acid), followed by permeabilization solution (75% ethanol, 0.025% Triton-X100) and staining buffer. Staining solution (0.1% cresyl violet in staining buffer) was applied for 20 min followed by differentiation of sections in 95% ethanol for 30 s and dehydration with isopropanol and xylene for 10 min each.

## Immunofluorescence staining

Immunofluorescent staining was performed for co-labeling of BrdU-positive cells with NeuN and S100β as described (*Steiner et al., 2008*). Briefly, sections were treated with 2 M HCl, washed extensively with PBS and blocked in PBS supplemented with 10% donkey serum and 0.2% Triton-X100 for 1 hr at room temperature, followed by incubation with primary antibodies overnight at 4°C (rat anti-BrdU 1:500, Serotec; mouse anti-NeuN 1:100, Merck Millipore; and rabbit anti-S100β 1:200, Abcam). Secondary antibodies were incubated for 4 hr at room temperature (anti-rat Alexa 488 1:500; anti-mouse Cy5 1:500; and anti-rabbit Cy3 1:500; all from Jackson ImmunoResearch). Nuclei were counterstained using 4′,6-diamidino-2-phenylindole (DAPI; 3.3 μg/ml) for 10 min. All washing steps were performed in PBS. Sections were mounted onto glass slides and cover-slipped using Aqua-Poly/Mount (Polysciences, Inc.). Imaging was performed with the ZEISS Apotome and the Software AxioVision software with optical sectioning mode. To determine total numbers of new-born neurons and astrocytes, 100 randomly selected BrdU immuno-positive cells along the rostro-caudal

axis of the dentate gyrus were investigated for co-expression with NeuN or S100β. The final numbers of surviving new neurons and astrocytes were obtained by multiplying the total number of BrdU-positive cells (as determined by peroxidase-based immunohistochemistry) by the ratio of NeuN/BrdU-positive cells and S100β/BrdU-positive cells.

## Brain morphometry and volumetry

The mossy fiber (MF) projections are characterized by a high content of the presynaptic vesicle protein synaptoporin (*Krebs et al., 2011*; *Singec et al., 2002*), therefore the volumes of the MF projections were estimated on sections immunolabeled against synaptoporin and counterstained with Nissl for a better distinction between neuronal cell layers. Volumetric analysis was performed on every sixth section with a semiautomated morphometric system consisting of a CCD camera (Hitachi) connected to a light microscope (Leica DM-RXE) using a 10x objective and the Stereoinvestigator 7 software (MBF Bioscience). Structures were overlaid with the Cavalieri estimator probe grid of 25 μm and every grid point belonging to the particular structure of interest was selected. Volume estimates were calculated in the software taking into account the sampling interval (240 μm) and the section thickness (40 μm).

For the analysis of the cortex thickness, the areas of motor, entorhinal and cingulate cortices were defined as described by Diamond et al. (*Diamond et al., 1964*; *Diamond et al., 1985*). We used the following coordinates of bregma: motor cortex −1.06 to −1.46, entorhinal cortex −2.30 to −2.80, cingulate cortex 1.34 to 0.50. Two to three constitutive sections from each animal were analyzed. Sections were scanned with a slide scanner (Axio Scan.Z1, Zeiss, Germany) and measured using the ZEN blue software (Zeiss, Germany).

Sections from several animals had to be excluded because of insufficient staining quality or damage to the tissue in the respective areas: hippocampus volumetry, 1 CTRL, 1 ENR mouse; motor cortex, 2 CTRL, 2 ENR mice; entorhinal cortex, 1 CTRL, 2 ENR mice; cingulate cortex, 3 CTRL mice.

## Analysis of blood samples

Blood was collected into EDTA-coated tubes (Sarstedt) from the abdominal cavity during the perfusion immediately after the right ventricle was opened. Blood samples were incubated for 1 to 2 hr at room temperature, and centrifuged at $2000 \times g$ for 15 min at 4°C. Plasma was centrifuged a second time and stored at −80°C. Plasma samples were assayed for glucose (Amplex red glucose/glucose oxidase assay kit, Invitrogen), cholesterol (Amplex red cholesterol assay kit, Invitrogen), triglycerides (Triglycerides colorimetric quantification kit, Abcam) and corticosterone (Corticosterone ELISA kit, Enzo) following the manufacturers' instructions. Log-logistic concentration curves were calculated from standards in R using the *drm* function from the *drc* package (*Ritz et al., 2015*). Corticosterone and triglyceride measures were log-transformed to normality.

## Statistics

All experiments were carried out with the experimenter blind to the experimental group. The data from this study have been deposited at Dryad (*Körholz et al., 2018*). Statistical analyses were carried out using the statistical software R (*R Core Team, 2014*). Data were tested for normality using the Shapiro-Wilk-test. For normally distributed measures, we used Welch's *t*-test to compare means and *F*-test to test for equality of variance between groups. For repeated measures (longitudinal data), a linear mixed regression was performed using the *lmer* function from the *lme4* package (*Bates et al., 2015*), and *p*-values were obtained by the likelihood ratio test of the full model against the model without the analyzed effects. For non-normal data, we performed the Wilcoxon rank sum test using the function *wilcox.test* as a non-parametric equivalent for the *t*-test, or the Brown-Forsythe test using the *leveneTest* function from the *car* package with the parameter *center* set to median as a more robust form of Levene's test to compare the variances between groups. Longitudinal non-normal or heteroscedastic data were analyzed using a rank-based non-parametric test using the *nparLD* function from the *nparLD* package, which reports a Wald-type test statistic for each of the effects and their interactions (*Noguchi et al., 2012*). All tests were two-tailed and differences were considered to be statistically significant at a $p < 0.05$. Data were visualized using the ggplot2 package (*Wickham, 2011*). In the box-whisker plots, center line and plus sign mark the median and mean, respectively. Upper and lower hinges indicate first and third quartiles. The upper whisker

extends from the hinge to the largest value no more than 1.5 times the interquartile range (IQR, a distance between the first and third quartiles); the lower whisker extends from the hinge to the smallest value at most 1.5 times IQR. Full results of statistical tests are available in *Supplementary file 2*.

## Acknowledgements

We thank Anne Karasinsky and Sandra Günther for taking care of our animals. We thank Alexander Garthe for his input while designing the behavioral experiments. We are grateful to all members of the Kempermann laboratory for assistance during the perfusion and collection of the various specimens.

## Additional information

### Funding

| Funder | Author |
|--------|--------|
| Deutsches Zentrum für Neuro-degenerative Erkrankungen | Gerd Kempermann |

The funders had no role in study design, data collection and interpretation, or the decision to submit the work for publication.

### Author contributions

Julia C Körholz, Formal analysis, Investigation, Visualization, Writing—original draft, Writing—review and editing; Sara Zocher, Formal analysis, Validation, Investigation, Methodology, Writing—original draft, Writing—review and editing; Anna N Grzyb, Data curation, Formal analysis, Visualization, Methodology, Writing—original draft, Writing—review and editing; Benjamin Morisse, Conceptualization, Investigation, Writing—review and editing; Alexandra Poetzsch, Investigation, Writing—review and editing; Fanny Ehret, Investigation, Methodology, Writing—review and editing; Christopher Schmied, Software, Investigation, Methodology, Writing—review and editing; Gerd Kempermann, Conceptualization, Supervision, Funding acquisition, Visualization, Methodology, Writing—original draft, Project administration, Writing—review and editing

### Author ORCIDs

Gerd Kempermann http://orcid.org/0000-0002-5304-4061

### Ethics

Animal experimentation: All experiments were conducted in accordance with the applicable European and national regulations (Tierschutzgesetz) and were approved by the responsible authority (Landesdirektion Sachsen), approval number 24-9168.11-1/2010-17.

### Decision letter and Author response

Decision letter https://doi.org/10.7554/eLife.35690.020
Author response https://doi.org/10.7554/eLife.35690.021

## Additional files

### Supplementary files

• Supplementary file 1. Phenotype data table. Abbreviations: NOR, novel object recognition test; OF, open field test; IMF, infrapyramidal mossy fibers; SMF, suprapyramidal mossy fibers; DG, dentate gyrus.
DOI: https://doi.org/10.7554/eLife.35690.014

• Supplementary file 2. Summary data, test statistics and *p*-values from statistical analyses of differences between CTRL and ENR mice. The *p*-values were not corrected for multiplicity of tests. For

longitudinal data, individual comparisons were made only when the omnibus test indicated significant difference between groups. In this case, both uncorrected and adjusted *p*-values are reported. Correction was performed using the Holm method (*Walsh and Cummins, 1979*) implemented using the *p.adjust* function in R. Abbreviations: LRT, likelihood ratio test; WTS, Wald-type statistic; NOR, novel object recognition test; OF, open field test; IMF, infrapyramidal mossy fibers; SMF, suprapyramidal mossy fibers; DG, dentate gyrus; st. dev., standard deviation.
DOI: https://doi.org/10.7554/eLife.35690.015

• Transparent reporting form
DOI: https://doi.org/10.7554/eLife.35690.016

## Data availability

All data generated or analysed during this study are included in the manuscript and supporting files and have been deposited at Dryad. Source data files have been provided for Figures 3 and 7.

The following dataset was generated:

| Author(s) | Year | Dataset title | Dataset URL | Database, license, and accessibility information |
| --- | --- | --- | --- | --- |
| Julia C Körholz, Sara Zocher, Anna N Grzyb, Benjamin Morisse, Alexandra Poetzsch, Fanny Ehret, Christopher Schmied, Gerd Kempermann | 2018 | Data from: Selective increases in inter-individual variability in response to environmental enrichment in female mice | http://dx.doi.org/10.5061/dryad.12cm083 | Available at Dryad Digital Repository under a CC0 Public Domain Dedication |

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
