## [Decision Letter]

Thank you for sending your article entitled "Selective increases in inter-individual variability in response to environmental enrichment" for peer review at *eLife*. Your article is being evaluated by Michael Frank as the Senior Editor, a Reviewing Editor, and three reviewer. The following individual involved in review of your submission has agreed to reveal her identity: Amelia J Eisch (Reviewer #1).

Summary:

The major conclusion of this basic research is that inbred isogenic female mice exposed to an enriched environment (ENR) for 3.5 months eventually show greater individuality on key measures of brain and behavior. As only a subset of measures show this increased variance, the authors interpret this as ENR driving the expression of individuality in these brain and behavior measures. This study probes beyond the authors' previous work into the experience-dependent biological basis of individuality using variance as a measure of individuality. The focus on differences in variance, rather than mean effects, is important and all too often ignored. This approach, broadly speaking, is likely to improve our understanding of experience-brain-behavior relationships (and, as is alluded to, is relevant for personalized medicine, for example). This is, overall, a comprehensive assessment of the effects of environmental enrichment on individual differences and a valuable analysis of the association of behavioral and biological variables and how the structure of those relationships is impacted differently by experience. We have questions in regards to statistical analyses, clarification of the ENR approach (its specificity, longevity, role in driving vs. revealing individuality), and comparison with prior work, and these are elaborated on in the "essential revisions" below.

Essential revisions:

In addition to the strengths of this work, several unresolved questions were raised by the reviewers. The essential revision requirements are listed below.

Given the emphasis on variance, issues relevant to statistics, data presentation, and animals used are extremely important to this work. For example, a significant strength of the work is the large number of behavioral and physiological variables examined. While the authors provide statistical methods in the main text and a complete table of statistical analyses in Supplementary file 1, the manuscript warrants discussion of how the overall experiment-wise error rate was controlled. In addition, reliance on p value magnitude as an index of strength of the difference should be either bolstered with more discussion or removed. The authors should also clarify exactly what the box and whisker plots represent, including the dots that appear above/below the bars. Indeed, individual data points should be provided in figures where possible. If symbol for statistical significance is indicated in figure, this should be large enough to be seen and also mentioned in legend. Currently asterisk is only in legend for Figure 7. Given the importance of strain to individuality research, in methods clarify whether these are indeed C57BL/6J (J=bred at Jackson Labs and shipped via Janvier) or a substrain that has established breeding Janvier. In text, avoid use of C57BL/6, since JAX states this does not exist (https://www.jax.org/news-and-insights/jax-blog/2016/june/there-is-no-such-thing-as-a-b6-mouse).

What is the specificity of the ENR effects? The authors state ENR "generally does not increase variability" but "is a [ed. not "the"] tool of choice to investigate the bases of individualization". The authors are encouraged to be clear in the discussion about whether they think ENR is a good tool for promoting individuality or not.

The authors say "…ENR is a suitable model to dissect processes leading to brain individualization". Rather than leading, driving, or causing individuality, the authors are encouraged to specifically discuss that ENR may reveal and/or amplify preexisting differences, such as those (particularly epigenetic differences) that may relate to gestation, early life, etc.

On a related note: Do the authors predict that other activity-dependent plasticity experiences – learning, social interaction, stress – or even short term experiences – disrupted light cycle – would also results in individualization and may also be used to "dissect processes leading to brain individualization", or is there something specific about ENR? In this case the presumed "individuality" would not be a long-term developmental phenotype (where "small initial differences are augmented through self-reinforcement"). Can the authors address this via discussion of their current or past work, or can this be addressed experimentally? What are the implications of this statement for other activity-dependent phenomena, like expression of immediate early genes?

The use of female mice raises two points. First, the authors should consider whether environmental effects on individualization interact with sex. Second, what is the authors' perspective on the stability or durability of this "expressed individuality" (greater variance) over time? This is important since the mice are (a) female and (b) relatively young, and (c) since studies suggest neurogenesis may not be a lifelong driver of hippocampal function. Therefore, brief speculative comments on sex-or age-induced "cementing", "inhibition", or "activity-independence" etc. of variance is warranted.

The issue of variance in general merits discussion. For example, a more specific summary or discussion of the negative findings (the many measures for which there is not increased variance after ENR) would is warranted, particularly since the measures taken range from relatively detailed high resolution (survival of new neurons) while others are more crude (body weight, organ weight). Negative results in measures of equal resolution (e.g. perhaps another cortical region aside from motor cortex) would help address this question, and also provide insight into the specificity of the ENT variance in exploration. Also, given the lack of variance in other indices of "fitness" in this study, it seems inappropriate to say there is an "increased range of fitness in ENR mice". Certainly, they are more fit based on liver, heart, cholesterol, etc., but there is no increased variance in these measures in ENR mice. Also, it is understandable the authors use a isogenic, inbred population since they are looking for divergence, or the ability of ENR to induce/reveal changes over time. As humans don't start as isogenic, inbred organisms, this inherent complication of the translational nature of this work merits additional discussion.

The authors discuss key differences with their prior work in regards to preference for novel object. However, the difference here with prior work in regards to body weight is not discussed (here ENR body weights less than controls, Freund et al., 2015 ENR body weights more than controls).

The authors report roaming entropy was found to be significantly lower in enriched versus control mice. This is a surprising finding given that the authors contend that roaming entropy drives brain plasticity. It is also hard to determine how this result relates to RE findings in the previous publications from this group. Again, this should be addressed in the Discussion section.

[Editors' note: further revisions were requested prior to acceptance, as described below.]

Thank you for resubmitting your work entitled "Selective increases in inter-individual variability in response to environmental enrichment in female mice" for further consideration at *eLife*. Your revised article has been favorably evaluated by Michael Frank (Senior Editor), a Reviewing Editor, and two reviewers.

The manuscript has been improved but there is one remaining issue that needs to be addressed before acceptance. Specifically, per reviewer 3's request, clarify in the Discussion section that the potential role of the relative strength of social hierarchies is speculative and state that sex differences in enrichment-induced individuation is unknown.

*Reviewer #2:*

The authors have done a suitable job of improving the manuscript in response to reviewer concerns. I have no further suggestions.

*Reviewer #3:*

The revised manuscript by Korholz et al., has, in general, adequately addressed the concerns this reviewer raised in the previous critique.

The first issue raised was the exclusive use of female mice in these experiments. The authors have now addressed this issue in the Discussion section. The authors reference previous work on sex differences in response to enrichment on several measures. These are not measures of individuation however. The potential role of the relative strength of social hierarchies is interesting but highly speculative. The authors need to provide a rationale for selecting female as opposed to male mice. The Discussion section also needs a clear statement that any sex differences in enrichment-induced individuation are unknown, and lack of accounting for sex differences represents a clear limitation of the current study.

The second issue raised was the concern of a potential confound associated with multiple statistical comparisons. The authors have also now addressed this issue in the Discussion section. The Introduction to where p values came from and what they mean is superfluous. This section does contain a cogent rationale for why an overall experiment-wise error rate was not used. This is responsive to the critique as there are dissenting opinions on the use of an experiment-wise error rate in all situations. What is important is a clear rationale for the authors' choice.

The authors have now addressed the inconsistency with their earlier results relative to the effect of enrichment on body weight. They have also addressed the surprising finding that RE was lower in enriched mice. This result is interpreted as a differential response to novelty. The authors should comment on the utility of using RE in a 5 min session and how this differs from the data subjected to RE in their previous papers.

---

## [Author Response]

Essential revisions:In addition to the strengths of this work, several unresolved questions were raised by the reviewers. The essential revision requirements are listed below.Given the emphasis on variance, issues relevant to statistics, data presentation, and animals used are extremely important to this work. For example, a significant strength of the work is the large number of behavioral and physiological variables examined. While the authors provide statistical methods in the main text and a complete table of statistical analyses in Supplementary file 1, the manuscript warrants discussion of how the overall experiment-wise error rate was controlled.

In the revised version we explained our statistical approach in more detail and discussed the issue of error rates and their consequences on our interpretation. In the Discussion section we included the following paragraph:

“To evaluate effects of ENR we used statistical tests appropriate for each given data distribution (as described in the Materials and methods) and interpreted *p* values smaller than 0.05 as a sufficient evidence for existing influence of ENR on a particular phenotype. *p* values were introduced by Ronald A. Fisher as an informal index to indicate whether or not the null hypothesis of no effect fails to account for the whole of the observations [summarized in Lehman 1993 – see at the end; Goodman 1993 – PMID: 8465801]. Lower *p* values imply lower likelihood of the null hypothesis given the obtained data, with strength of evidence being interpreted as weak to moderate in the range between 0.1 to 0.01, and strong to very strong at or below 0.001 (Goodman, 1999). The negative findings, though, are less clear to evaluate. With 40 individuals in each experimental group, 5% pre-set significance level enabled us to detect in a two-tailed variance test an effect size of 2.5 (ratio of variances) with power of 0.8. This is a relatively large effect and moderate power, which implies that smaller changes in variance, as well as some of the stronger effects, could have been missed. At the same time, we would argue that the identified effects are robust enough to be biologically relevant. Furthermore, we adopted marginal, that is individual, interpretation of responses to ENR of each of the assayed phenotypes and did not control for an experiment-wide type I error rate. Correction of *p* values upon multiplicity of tests stems from posing the universal null hypothesis that no association exists between any pair of variables under investigation (Rothman, 1990). The question we posed was, however, not whether ENR influences variance or mean of *any* trait (a universal null hypothesis), in which case the control of experimental error rate would be necessary, but rather each of the specific responses was of interest (for the distinction of various scenarios see Cook and Farewell 1996). Most importantly, however, the correction for multiple tests leads to inflation of type II errors (false negatives) and hence introduces deleterious for statistical inference “penalty for peeking”, i.e. the more parameters are investigated, the less likely each of the true associations is to be detected (Rothman 1990; Perneger 1998). Significant *p* values in our experiment were not uniformly distributed across all phenotypes, supporting our view that we achieved a balance between false negative and false positive error rates sufficient to provide a coherent overview of effects of ENR on a wide scope of traits.”

In addition, reliance on p value magnitude as an index of strength of the difference should be either bolstered with more discussion or removed.

We discussed the general issue (see above), but otherwise followed the suggestion to remove the points in question. The introduced changes do not affect the overall interpretation.

The authors should also clarify exactly what the box and whisker plots represent, including the dots that appear above/below the bars. Indeed, individual data points should be provided in figures where possible.

We included the description of the box and whisker plot in the Materials and methods section, as well as in the first figure where they appear. We also redrew the figures to show all data points. In the box-whisker plots, as introduced by Tukey, (1977), upper and lower hinges indicate first and third quartiles. The upper whisker extends from the hinge to the largest value no further than 1.5 times the interquartile range (IQR, a distance between the first and third quartiles); the lower whisker extends from the hinge to the smallest value at most 1.5 times IQR. Data beyond the end of the whiskers are called “outlying” points and are plotted individually.

If symbol for statistical significance is indicated in figure, this should be large enough to be seen and also mentioned in legend. Currently asterisk is only in legend for Figure 7.

We included the explanation in each figure legend and improved readability.

Given the importance of strain to individuality research, in methods clarify whether these are indeed C57BL/6J (J=bred at Jackson Labs and shipped via Janvier) or a substrain that has established breeding Janvier. In text, avoid use of C57BL/6, since JAX states this does not exist (https://www.jax.org/news-and-insights/jax-blog/2016/june/there-is-no-such-thing-as-a-b6-mouse).

We indeed used C57BL/6JRj, bred by JAX and purchased from Janvier. We corrected and specified the related information in Materials and methods section, and in the Introduction. Thank you for pointing this out.

What is the specificity of the ENR effects? The authors state ENR "generally does not increase variability" but "is a [ed. not "the"] tool of choice to investigate the bases of individualization". The authors are encouraged to be clear in the discussion about whether they think ENR is a good tool for promoting individuality or not.

This study shows that the increased variability in response to ENR is specific to adult hippocampal neurogenesis, motor cortex thickness and exploratory behavior. Thus, ENR is a decent tool to promote experience-dependent individuality in structural brain plasticity. Whether ENR is the only behavioral paradigm that induces individuality in these parameters, is currently unknown and needs further investigation. To clarify these points, we emphasized the specificity of the variance-inducing effects of ENR and specified that ENR is a suitable tool to study mechanisms of experience-dependent brain individualization. We revised the text in the Discussion section:

“In conclusion, ENR does not generally increase variability across all domains. ENR-induced increases in variance were specific to exploratory behavior, adult neurogenesis and motor cortex thickness. The correlation pattern of these parameters with other traits was complex, with ENR remodeling many of the associations. We do not think, however, that increased structural variability is limited only to neurogenesis and motor cortex, but that induced changes are very specific and can be revealed only if appropriate aspects of plasticity are examined. ENR arises from this study as a more holistic paradigm than often assumed and proves to be a decent tool to investigate the bases of experience-dependent brain individualization.”

The authors say "…..ENR is a suitable model to dissect processes leading to brain individualization". Rather than leading, driving, or causing individuality, the authors are encouraged to specifically discuss that ENR may reveal and/or amplify preexisting differences, such as those (particularly epigenetic differences) that may relate to gestation, early life, etc.

We included a section discussing this comment in the Discussion section of the manuscript:

“We hypothesize that increases in variability are a result of the progressive amplification of initially small inter-individual differences that existed before the start of the experiment, or that were introduced by stochastic events in the initial period of ENR housing. Potential sources of pre-existing variation include prenatal influences on the pregnant mother, intrauterine positioning of the fetus as well as early postnatal experiences (Saal, 1981; Lathe, 2004). Especially early life-experiences are known to change epigenetic modifications in the brain, which contribute to the long-term control of gene expression. For instance, differences in maternal care in rats led to differences in the DNA methylation state of the glucocorticoid receptor in the hippocampus in the offspring (Weaver et al., 2004). Moreover, human twin studies have suggested that monozygotic twins increasingly differ in epigenetic marks from early life to adulthood, presumably as a result of the different experiences they make (Fraga et al., 2005; Cheung et al., 2018). Environmental enrichment builds on the initial variation and amplifies the differences by providing opportunity for the development of individual behavior.“

On a related note: Do the authors predict that other activity-dependent plasticity experiences – learning, social interaction, stress – or even short term experiences – disrupted light cycle – would also results in individualization and may also be used to "dissect processes leading to brain individualization", or is there something specific about ENR? In this case the presumed "individuality" would not be a long-term developmental phenotype (where "small initial differences are augmented through self-reinforcement"). Can the authors address this via discussion of their current or past work, or can this be addressed experimentally? What are the implications of this statement for other activity-dependent phenomena, like expression of immediate early genes?

This is an important and interesting point. The question of specificity itself is a classical question of the enriched environment literature. Nevertheless, published experiments tend not to provide information that would allow addressing that point and have too small group sizes to allow clear conclusions about such individualizing effects. This discrepant situation has been one of the motivating factors for the current study. We have ongoing efforts in the lab to provide such analyses from the literature and our own previous data sets but despite their relatedness to the mentioned question, they leave the framework of the current study. We commented the issue in the revised Discussion section:

“The question arises whether ENR is unique among activity-dependent plasticity experiences in inducing behavioral and structural divergence. Published studies tend not to provide sufficient information about individualizing effects of other manipulations due to modest group sizes. ENR is a complex paradigm, in which inanimate aspects of the environment and social interactions intertwine over prolonged periods of time. We would hold the view that both this complexity and duration are essential elements in consolidation of the induced changes. Accordingly, in our previous study (Freund et al., 2013; Freund et al., 2015), the patterns of general activity, RE and both social and non-social behaviors recorded towards the end of the ENR exposure could not be predicted by the initial differences between animals.”

The use of female mice raises two points. First, the authors should consider whether environmental effects on individualization interact with sex.

This is a very important point. We included a discussion of this point in the Discussion section of the manuscript:

“In this study, ENR increased variation within a group of female mice. Male mice are known to similarly respond to ENR with increased hippocampal neurogenesis (Zhang et al., 2018). In contrast, several studies reported sex differences in behavioral responses towards ENR, with female mice being more susceptible to the positive effects of ENR on cognition (Hendershott et al., 2006; Wood et al., 2010; Laurence Coutellier and Würbel, 2009). Since male mice build stronger hierarchies than females, we expect that the contribution of the social interaction on individuality development is stronger in a male group compared to a female group of mice. Dominant males influence the behavior and stress levels of subordinate males (Curley, 2016), which could lead to an even stronger and faster individualization in ENR that is less instructed by activity-dependent brain plasticity. On the other hand, increased social distress in subordinate animals might blur individualization effects.”

Second, what is the authors' perspective on the stability or durability of this "expressed individuality" (greater variance) over time? This is important since the mice are (a) female and (b) relatively young, and (c) since studies suggest neurogenesis may not be a lifelong driver of hippocampal function. Therefore, brief speculative comments on sex-or age-induced "cementing", "inhibition", or "activity-independence" etc. of variance is warranted.

We included a following section discussing this comment in the Discussion section of the manuscript:

“According to our hypothesis of positive feedback through experience, the individualization that occurs in the first months of ENR housing should lead to a permanent discordance of behavior, responses to cognitive challenges, and possibly also brain morphology. In support of this, we have previously shown that mice establish stable behavioral trajectories in the first two months of ENR housing that are maintained for the time of monitoring (Freund et al., 2013) and are presumably kept up for long-term. Whether life-long ENR housing would increase the inter-individual variability even more over time and whether the inter-individual differences are stable after withdrawal of the ENR stimulus is a subject of current investigations. While the positive influence of ENR on neurogenesis and behavior is independent of age (Kempermann et al., 1998), housing of mice for longer than 3 months in ENR does not further increase experience-dependent neurogenesis (Kempermann and Gage, 1999). Concurrently, exploratory activity of mice was shown to decrease with time in ENR (Freund et al., 2013), suggesting, together with the age-related decline in neurogenesis, that the strength of the iterative feedback between neurogenesis and behavior would decrease, which could result in a plateauing of the individuality effect with time.”

The issue of variance in general merits discussion. For example, a more specific summary or discussion of the negative findings (the many measures for which there is not increased variance after ENR) would is warranted, particularly since the measures taken range from relatively detailed high resolution (survival of new neurons) while others are more crude (body weight, organ weight). Negative results in measures of equal resolution (e.g. perhaps another cortical region aside from motor cortex) would help address this question, and also provide insight into the specificity of the ENT variance in exploration.

This is an important point, which we discussed along with the control of the experiment-wise error rates in the Discussion section. In our study, 40 individuals in each experimental group allowed us to detect an effect size (ratio of variances) for a two-tailed variance test of 2.5 with the power of 0.8. This is a relatively large effect, which implies that smaller changes in variance could be missed, but at the same time we would also argue that the identified effects are robust enough to be biologically relevant. From the statistical point of view, false negative findings are most likely to arise for traits with low precision of measurement or otherwise with high relative variance. Nonetheless, we were able to observe significant effects on variance in phenotypes with a relatively high coefficient of variance, such as neurogenesis or in behavioral tasks.

In addition to neurogenesis, we assayed several parameters that are indicative of brain plasticity: volumes of mossy fiber projections and thickness of three different cortical areas (cingulate, entorhinal and motor cortices), out of which large increase in variance could only be observed in motor cortex thickness. We do not think, however, that increased individuality is limited only to neurogenesis and motor cortex, but that induced changes are very specific and can be revealed only if appropriate aspects of plasticity are investigated. We introduced this point in the Discussion section. Currently, we initiated studies in new cohorts of mice aiming at finding other facets of brain plasticity which would undergo such induction of variance, but these are beyond the scope of the current report.

Also, given the lack of variance in other indices of "fitness" in this study, it seems inappropriate to say there is an "increased range of fitness in ENR mice". Certainly, they are more fit based on liver, heart, cholesterol, etc., but there is no increased variance in these measures in ENR mice.

This is correct, and we revised the text accordingly.

Also, it is understandable the authors use an isogenic, inbred population since they are looking for divergence, or the ability of ENR to induce/reveal changes over time. As humans don't start as isogenic, inbred organisms, this inherent complication of the translational nature of this work merits additional discussion.

We revised the last paragraph of the Discussion section:

“(…) In the laboratory setting, animals are relieved from pressures present in nature and therefore they are free to choose the degree of interaction with the environment. In our previous longitudinal study, we made the case that in a situation, in which both genes and (nominal) environment are kept constant, individuality emerges as a consequence of the so-called “non-shared environment”, i.e. the individual response to that environment and activity (Loseva, Yuan and Karnup, 2011). This situation is comparable with monozygotic twins, which—even when raised in the same household—develop differences in behavior, appearance and disease susceptibility over time. The underlying mechanisms that drive this divergence are, however, unknown and difficult to address in human studies. Here, we present an animal model to study the influence of the non-shared environment on individualization and its relation to brain plasticity. Our data suggests that multivariate studies with a large number of individuals and, ideally, a longitudinal design are needed to elucidate the exact contribution of the non-shared environment to the overall outcome of increased individualization. In perspective, the model of long-term ENR can be extended to analyze the development of individuality in a genetically variable population to provide insights into the interaction of genes with the non-shared environment.”

The authors discuss key differences with their prior work in regards to preference for novel object. However, the difference here with prior work in regards to body weight is not discussed (here ENR body weights less than controls, Freund et al., 2015 ENR body weights more than controls).

In our current laboratory, we commonly observe decreased body weight of mice after ENR. We have also observed a lower body weight in our old studies (for example: Kempermann and Gage, 1999). The experiment described in two papers, Freund et al., 2013, and Freund et al., 2015, involved two cohorts of 40 mice and a small control group of 15 mice. In comparison to this control group, one ENR cohort did not show change in body weight (Freund et al., 2013), while the second showed only a trend towards increased body weights (p = 0.09), therefore we would not overemphasize this single observation. It has been shown that local conditions between animal facilities influence physiological parameters, such as weight, and behavior (ref. Crabbe et al. 1999) and is thus possible that also response to ENR in activity, which will influence both lean mass and energy expenditure, or feeding are specific to the location. Given the small control group in the previous experiment we could not conclude if the observed lack of differences in variance would represent a general pattern: such questions were exactly the starting point for the current study. Because of the substantial length of the revised discussion and the fact that this contradictory finding does not bear significance for drawing overall conclusions from the presented work, we only briefly pointed to the discrepancy by including the following sentence in the Discussion section:

“[…] In the present study, we also observed decreases in body and liver weights, as well as lowering of plasma cholesterol, which indicates that ENR alone has a moderate beneficial effect on metabolism even in the absence of intense physical exercise. Although we routinely record lower body weights in mice upon ENR (Kempermann and Gage, 1999; unpublished observations), this response might be subjected to the local conditions between animal facilities (Crabbe et al. 1999), as in our previous experiment such an effect was not observed (Freund et al. 2013; Freund et al. 2015).”

The authors report roaming entropy was found to be significantly lower in enriched versus control mice. This is a surprising finding given that the authors contend that roaming entropy drives brain plasticity. It is also hard to determine how this result relates to RE findings in the previous publications from this group. Again, this should be addressed in the Discussion section.

We had this point hidden in the Discussion section (below), and now further elaborated on it.

“The behavior in the familiar environment of the ENR cage could have been influenced by the group interactions (Shemesh et al., 2013), while the data presented here reflects the individual response of the animals to novel situations and might, therefore, constitute a different construct.”

Accordingly, we modified the Discussion section:

“(…) In contrast to the study by Freund et al., where 21% of variance in adult neurogenesis could be explained by differences in cumulative roaming entropy (RE), an aggregated measure of the longitudinal behavioral trajectory, the current analysis detected no correlation between the number of newborn neurons and any of the cross-sectionally measured behaviors, including RE measured in the OF test. RE is a convenient single parameter describing the uniformity of coverage of a given space (Freund 2013; Freund 2015). Here we used RE to describe the exploration in the OF, because it carries more information compared to the traditional measures. Specifically, it does not rely on hard boundaries, such as the periphery and center of the open field, and also takes into account exploration within each of these zones (for example, a mouse which remains in the corner of an OF arena and a mouse that visits the entire perimeter of an arena might have spent the same amount of time in the periphery but differ in the RE measure). In the present study, RE in the OF was lower among ENR mice compared to CTRL. The behavior in the familiar environment of the ENR cage, however, is presumably driven by other factors and can be influenced by the group interactions (Shemesh et al., 2013), while the OF test reflects the individual response of animals to novel situations and might, therefore, constitute a different construct. Perception of novelty and incentive to explore an empty OF arena is likely to be different between CTRL animals, which spent their entire life in small cages, and the ENR group.”

[Editors' note: further revisions were requested prior to acceptance, as described below.]

Reviewer #3:

The revised manuscript by Korholz et al., has, in general, adequately addressed the concerns this reviewer raised in the previous critique.The first issue raised was the exclusive use of female mice in these experiments. The authors have now addressed this issue in the Discussion section. The authors reference previous work on sex differences in response to enrichment on several measures. These are not measures of individuation however. The potential role of the relative strength of social hierarchies is interesting but highly speculative. The authors need to provide a rationale for selecting female as opposed to male mice. The Discussion section also needs a clear statement that any sex differences in enrichment-induced individuation are unknown, and lack of accounting for sex differences represents a clear limitation of the current study.

To address this comment, we included the following changes in the text in the Discussion section of the second revised version:

“In this study, ENR increased variation within a group of female mice. We have used females to avoid the inter-animal conflict behavior that unrelated males show when put together at the delivery age of 4 weeks and to build on our previous ENR experiments (Kempermann and Gage, 1999; Freund et al., 2013; 2015). Male mice are known to similarly respond to ENR with increased hippocampal neurogenesis (Zhang et al., 2018). In contrast, several studies reported sex differences in behavioral responses towards ENR, with female mice being more susceptible to the positive effects of ENR on cognition (Coutellier and Würbel, 2009; Hendershott et al., 2016; Wood et al., 2010). Since male mice build stronger hierarchies than females, we expect that the contribution of the social interaction on individuality development is stronger in a male group compared to a female group of mice. Dominant males influence the behavior and stress levels of subordinate males (Curley, 2016), which could lead to an even stronger and faster individualization in ENR that is less instructed by activity-dependent brain plasticity. On the other hand, increased social distress in subordinate animals might blur individualization effects. However, whether ENR housing leads to the development of inter-individual differences in behavior and brain plasticity also in male mice, is currently unknown and should be addressed in future experiments.”